# The Nutritional Gene Expression Regulation Potential of a Lysolecithin-Based Product

**DOI:** 10.3390/cimb47070548

**Published:** 2025-07-15

**Authors:** Veerle Van Hoeck, Riet Spaepen, Bart Forier

**Affiliations:** Kemin Europa N.V., 2200 Herentals, Belgium; riet.spaepen@kemin.com (R.S.); bart.forier@kemin.com (B.F.)

**Keywords:** lysolecithin, Caco-2 cells, nutritional regulation of gene expression, phospholipids, lysophospholipids

## Abstract

Lysolecithin is a performance-enhancing product for livestock. Lysolecithins contain functional phospholipids (PLs) and lysophospholipids (LPLs) and have been used in monogastric feed formulations because they can enhance lipid emulsification, digestion, and absorption (surface chemistry). Another underexplored aspect is that lysolecithin mixtures can serve as signaling via so-called nutritional gene expression-regulating action. The scope of this study was to fully understand the potential of a lysolecithin source derived from soybeans to influence intestinal nutrient transport in the intestinal tract. In this context, in vitro cell culture data with intestinal Caco-2 cells revealed that a lysolecithin-based product can significantly improve intestinal cell viability. Furthermore, a Transwell culture experiment showed that lysolecithins can significantly trigger gene expression. The most significantly affected genes could be correlated with G-coupled protein cascades. Enrichment analyses showed that amino acid transport and lipid metabolism pathways are significantly affected. Furthermore, the polarized cell culture revealed that the studied lysolecithin could affect the abundance of metabolites/nutrients in the basolateral compartment when applied apically, indicating that its action exceeds surface chemistry. In conclusion, the data on intestinal cell viability, gene expression, and metabolite abundance seem to reveal the bioactivities of lysolecithin. The latter data suggest that the specific lysolecithin source used here is more than a biosurfactant; more specifically, it seems to be a potent bioactive mixture of amphiphilic compounds triggering cell signaling pathways.

## 1. Introduction

Lysolecithin provides value to the poultry and swine industry, either complimentary to the existing diet or in reformulation; it substantially supports growth performance and feed efficiency, as well as providing a multitude of additional benefits across all stages of production [1,2,3,4].

One major effect of lysolecithins is the digestion of fats and oils in broilers, which is a complex process [1] that involves emulsification, hydrolysis, and absorption of the ingested fats [5]. Several factors can affect the fat digestion process by interfering with one or more of these steps. In order to promote the digestion and absorption of nutrients—especially of dietary fats and oils—lysolecithins are supplemented to the diets of farm animals [6]. It is postulated that, together with bile salts, the LPLs act as an emulsifier within the first stages of lipid digestion [7]. Lysolecithins function as effective surface-active agents for emulsifying fat [8]; they enable the formation of smaller fat droplets, increasing the contact surface for the endogenous enzyme lipase to hydrolyze the triglycerides into monoglycerides and free fatty acids [8].

However, there seems to be more than just a surface chemistry effect of lysolecithins. LPLs are key components of all biological membranes, but they also have a variety of other functions, such as cellular messaging and enzyme activation [9]. In this context, it has been shown that lysolecithin can alter intestinal membrane fluidity and protein channel formation across the enterocyte membrane [4]; it can also stimulate gene expression associated with collagen deposition, which enhances villus length, gut integrity and strength [10], and intestinal absorption surface area via proliferation and differentiating pathways [10]. The latter events can most likely be explained by nutritionally induced changes [11].

Furthermore, these improvements in gut structure and function could explain the better utilization of the available dietary nutrients, including energy [4,12,13] and protein [7,14], which may subsequently drive the improvements in performance efficiency and carcass yields reported in monogastric animals by Li et al. [10].

Phospholipids and lysophospholipids are bioactive lipids and can signal through G-protein-coupled receptors (GPCRs). The best-studied lysophospholipids are lysophosphatidic acid (LPA) and sphingosine 1-phosphate (S1P). The mechanisms of lysophospholipid recognition by an active GPCR, along with the activation of lysophospholipid GPCR–G–protein complexes, remain unclear [15,16,17].

The nutritional gene-regulating properties of PLs and LPLs depend on the fatty acid (FA) chain length, the saturation rate, and the functional moiety attached to the glycerol backbone [18,19,20,21,22]. However, considering these three variables, several questions still arise regarding the gene regulation potency of lysolecithins, especially in the intestinal tract. For example, Brautigan et al. [23] showed that lysophosphatidylcholine (LPC), which is a fraction of the lysophospholipids, did not affect the intestinal tract as much as the entire lysolecithin supplement. However, one needs to question the characteristics of the LPCs that were tested in this study. An increasing number of studies reveal that it is not the inclusion level of LPLs that is important, but rather the quality of the LPLs, which can be defined by FA characteristics. For example, LPC may have inflammatory properties that appear to be highly dependent on the fatty acyl chain tail [24]. As reviewed by Drzazga et al. [24], pro-inflammatory properties have been mainly associated with LPC containing saturated FAs on the acyl tail, especially C14:0 and C16:0, whereas anti-inflammatory properties have been observed with LPC containing (poly)unsaturated FAs, especially 20:4 and 22:6. Lysophosphatidylethanolamine (LPE) has been shown to express similar inflammatory properties to those found for LPC [25].

However, many of these functions are still poorly understood. Additionally, as many of the properties ascribed to PL and LPLs are highly dependent on their specific composition (e.g., fatty acyl chain), different PL and LPLs may behave and interact differently.

Previous research has highlighted that lysolecithins directly enhance all phases of lipid digestion and FA absorption through a multi-level, synergistic mode of action [2,3,4]; however, there are indications that they affect more pathways influencing the digestion processes and absorption of other essential nutrients, such as amino acids, which have not yet been fully explored and unraveled. In this context, a soybean lysolecithin was fully characterized and its effect on intestinal cells was studied with a polarized/Transwell cell culture setup via a triple approach. More specifically, its effects on intestinal cell viability, gene expression, and metabolite transport were screened, according to the chart presented in Figure 1. Hereby, it was clarified that the effects of the lysolecithins exceed surface chemistry; as such, lysolecithins contain compounds that seem to have very potent signaling functions. This would imply that very small amounts of the right lysolecithins can have a significant impact on the intestinal tract metabolism and nutrient transport actions. This study aims to assess whether a soybean-derived lysolecithin affects intestinal epithelial cell viability, gene expression, and metabolite transport in vitro using a triple-assay approach.

## 2. Materials and Methods

### 2.1. Lysolecithin Characterization

The source of lysolecithin is soybean. In order to obtain lysolecithin, soybean-sourced lecithin has been treated with phospholipase. All analyses were performed at the ITERG (French institute specialized in vegetable oils and proteins, France).

Preparation and analysis of the methyl esters of fatty acids were carried out by gas chromatography to determine their composition and contents. Briefly, the total fatty acids were trans-methylated according to the acid-catalyzed method ISO 12966-2 [26] and analyzed on a gas chromatograph equipped with a flame ionization detector, according to the French standard NF EN ISO 12966-4 [27]. Individual FAMEs were identified by comparing their retention times with those of authentic standards eluted under the same conditions (Sigma Chemical Company, St. Louis, MO, USA). The results were expressed as a percentage of the total fatty acids. The FAMEs of total lipids were quantified using an internal standard.

The phospholipid contents were determined by using high-performance liquid chromatography (HPLC), as well as an evaporative light-scattering detector (SEDEX LT-ELSD SOLT, HPLC DDL Sedere, Thermo Fisher Scientific, Waltham, MA, USA) [26]. The different classes of phospholipids were separated using a silica column with separation parameters adapted from Becart et al. [28]. Identification of phospholipids and lysophospholipids was carried out by comparison with the retention time of pure standards (Avanti polar Lipids, Alabaster, AL, USA). Standard calibration curves for each phospholipid were used for their respective quantification in the lysolecithin products. The results were expressed as the g/100 g for each phospholipid. Results were analyzed using Chromeleon software (version 7.3.2; Thermo Fisher Scientific).

The sFA distribution over the LPL fractions was determined as follows: phospholipid classes were separated by thin-layer chromatography (glass plates 20 × 20 cm pre-coated with silica gel 60H), with migration parameters adapted from Weerheim et al. [29]. After migration and detection with 2,7-dichlorofluorescein, the silica gel areas corresponding to each phospholipid class were visualized under UV light, removed from the TLC plate, to be then derived into FAME by transmethylation method, as described in a previous study [30]. The resulting FA methyl esters (FAME) were analyzed by GC-FID [30], (GC, TRACE GC, Thermo Scientific, Waltham, MA, USA), equipped with a flame ionization detector (FID) and a split injector. A fused-silica capillary column (BPX 70, 60 m × 0.25 mm i.d., 0.25 μm film; SGE, Illkirch, France) was used with hydrogen as a carrier gas (inlet pressure: 120 kPa). GC peaks were integrated using Chromquest software5.0 (Thermofinnigan, Courtaboeuf, France).

### 2.2. Experimental Design

The human intestinal Caco-2 cell line was originally obtained from Prof. Alain Zweibaum (INSERM, Villejuif, France). To prepare the initial low-density (LD) cell stock, Caco-2 cells were seeded at 4.5 × 10^3^ cells/cm^2^ and sub-cultured at 50% confluence (5.4 × 10^4^ cells/cm^2^) for 10 passages, changing the medium every two days. A large stock of LD cells was produced and stored in liquid nitrogen (see the cell freezing protocol below), allowing all of the experiments to be performed within a range of four passages. In our experiment, Caco-2 cells were sub-cultured at low density as previously described in [31]. In vitro polarized cell culture setups (Transwell^®^ assay; Corning, Bologne Billancourt, France) with intestinal Caco-2 cells were cultured according to standard conditions on 6-well Transwell^®^ filters and filter-based inserts for approximately 21 days, in order to allow differentiation and polarization to confluent monolayers [31,32]. Briefly, cells were routinely sub-cultured at 50% density and were maintained at 37 °C in a 90% air–10% CO_2_ atmosphere in Dulbecco’s minimum essential medium (DMEM) containing 25 mM glucose, 3.7 g/L NaHCO_3_, 4 mM l-glutamine, 1% non-essential amino acids, 100 U/L penicillin, and 100 μg/L streptomycin (complete medium), supplemented with 10% heat-inactivated fetal bovine serum (FBS) (HyClone Laboratories, Logan, UT, USA). All reagents were obtained from Sigma unless otherwise stated. Two hours prior to exposure, the medium of the basal compartment was replaced with PBS containing Ca^2+^ and Mg^2+^ to allow for equilibration to the nutrient-depleted conditions at the serosal side of the cells. Similarly, the medium in the apical compartment was replaced with serum-free DMEM. Next, the PBS in the basal compartment was replaced with fresh PBS, and the cells were either treated or not with 0.5% (by weight per volume) lysolecithin (the soybean-derived lysolecithin source present in LYSOFORTE^®^) solubilized in serum-free DMEM, with 6 repeats per treatment. Before and after incubation, a quality control (QC) of the cell monolayer’s intactness was performed by measuring the transepithelial electrical resistance (TEER), using the automated REMS device. TEER is a non-invasive technique that measures the impedance between the lumen and the basolateral tissue. TEER measurements use a constant direct current applied by two electrodes: one connected with the lumen side, and the other one with the basolateral side. By applying Ohm’s law, it is possible to measure the related cell’s resistance [33]. The experimental setup is presented in Figure 2. The human intestinal Caco-2 cell line differentiates spontaneously in culture without supplementation of differentiating factors, and it has been extensively used as a model of the intestinal barrier for in vitro toxicology studies [34].

Note: The in vivo addition of lysolecithin in broiler diets is 250 g. A young broiler chicken eats 200 g of feed per day. This implies that the broiler will have an intake of ±0.5 g lysolecithin per day. The content of intestinal lumen in the broiler is estimated to be 100 mL. So, that means 0.5 g/100 mL = 0.5% exposure of lysolecithin in the intestinal tract per day. Hence, the dosage set up in this in vitro work is physiologically relevant for the in vivo situation.

### 2.3. Toxicity Screening

To improve the success of the Transwell^®^ experiment, we aimed to apply the highest dose without disturbing the barrier integrity and inducing strong toxicity. To this end, Caco-2 cells were cultured according to standard conditions in 96-well plates and allowed to differentiate for 14 days. Next, 0, 0.1, 0.5, 1, and 2% lysolecithin was added to serum-free exposure medium, filter-sterilized, and applied to the cells for 24 h. Next, the cell monolayer integrity was checked microscopically, and an MTT assay was used to measure possible toxic effects. Statistical procedures were carried out in JMP^®^, Version 15.0.0 (SAS Institute Inc., Cary, NC, USA, 1989–2019). The effects of treatments on variable parameters were analyzed via ANOVA. Post-hoc comparisons between treatments were carried out using Duncan’s test.

#### Evaluation of the Effects of Lysolecithins on Intestinal Cell Viability

The exposure was carried out over a 5 h period. After 5 h, the medium was discarded, the cells were washed twice with PBS, and then they were tested for cell viability using the MTT test with spectrophotometry. The MTT test measures mitochondrial respiration by the conversion of the yellow MTT start product ((3-(4,5-dimethylthiazol-2-yl)-2,5-diphenyltetrazolium bromide) to formazan by NAD(P)H-dependent oxidoreductase enzymes in viable cells. The cells were treated with 20 µL of MTT solution (5 mg/mL) in 100 µL of serum-free medium for 2 h. Next, the medium was removed, and the formazan crystals were dissolved in DMSO. Absorbance was measured using a plate reader at a wavelength of 570 nm. Statistical procedures were carried out in JMP^®^, Version 15.0.0 (SAS Institute Inc., Cary, NC, USA, 1989–2019). The effects of treatments on variable parameters were analyzed via student *t*-test.

### 2.4. Studying the Effects of Lysolecithins on Intestinal Gene Expression and Metabolite Transport/Production

Exposure was carried out for 5 h. The cells were scraped from the filter surface, pipetted into a DNA/RNA-free low-binding Eppendorf tube, and centrifuged at 3000 rpm for 1 min at 4 °C. The supernatant PBS was removed, and the cell pellet was frozen at −80 °C. The basal PBS was collected in clean glass tubes and immediately frozen at −80 °C. Next, the apical medium was removed, the cells were washed twice with PBS containing Ca^2+^ and Mg^2+^, and 1 mL of the same PBS was added to the cells to prevent them from drying out.

### 2.5. RNA Sequencing

The cells were stored at −80 °C prior to shipment to Novogene (Cambridge, UK), where the RNA-Seq was performed. In this experiment, cells from two wells were pooled together to retrieve enough biological material to extract the mRNA. Furthermore, 6 repeats were performed for each treatment, so a total of 12 samples were sent to Novogene.

#### 2.5.1. Total RNA Extraction

Frozen Caco-2 cell samples were immediately mixed with buffer from the RNeasy Mini columns kit (Qiagen, Hilden Geschäftsführer, Germany), and further RNA extractions were performed using a commercial purification kit (Qiagen, Hilden Geschäftsführer, Germany), according to the manufacturer’s recommendations.

#### 2.5.2. Library Preparation and Sequencing

From the RNA sample to the final data, each step—including sample testing, library preparation, and sequencing—influences the quality of the data, and the data quality directly impacts the analysis results. To guarantee the reliability of the data, QC was performed at each step of the procedure. The RNA-Seq workflow is depicted in Figure 3.

#### 2.5.3. Total RNA Sample QC

All samples needed to pass through the following three steps before library construction:(1)NanoDrop (ThermoFisher Scientific, Waltham, MA, USA): tests RNA purity (OD260/OD280);(2)Agarose gel electrophoresis: tests RNA degradation and potential contamination;(3)Agilent 2100 (Agilent Technologies, Digem, Belgium): checks RNA integrity.

The RIN values of the samples were 9.3, 9.4, 9.3, 9, 8.5, 9.1, 6.1, 9.2, 9.2, 9.3, 9 and 5 for the first six control samples and then the last six lysolecithin samples, respectively.

#### 2.5.4. Data Quality Control

Raw data (raw reads) of fastq format were firstly processed through fastp software (https://github.com/OpenGene, accessed on 3 May 2025). In this step, clean data (clean reads) were obtained by removing reads containing adapter, reads containing ploy-N and low-quality reads from raw data. At the same time, Q20, Q30 and GC content the clean data were calculated. All the downstream analyses were based on the clean data with high quality.

#### 2.5.5. Readsmapping to the Reference Genome

Reference genome and gene model annotation files were downloaded. Index of the reference genome was built using Hisat2 v2.0.5 and paired-end clean 1 reads were aligned to the reference genome using Hisat2 v2.0.5. Hisat2 [35] was selected as the mapping tool for that Hisat2 can generate a database of splice junctions based on the gene model annotation file and thus a better mapping result than other non-splice mapping tools. See Appendix A for mapping result.

#### 2.5.6. Quantification of Gene Expression Level

FeatureCounts [36] v1.5.0-p3 was used to count the reads numbers mapped to each gene. And then the FPKM of each gene was calculated based on the length of the gene and reads count mapped to this gene.

#### 2.5.7. Differential Expression Analysis

After the gene expression is quantified, statistical analysis of the expression data is required to screen the genes whose expression levels are significantly different in different conditions. The differential analysis is mainly divided in three steps: First the raw read count is normalized (DESeq method) mainly to correct sequencing depth. Next, the statistical model (Negative Binomial Distribution) is used to calculate the hypothesis test’s probability (*p*-value). Finally, multiple hypothesis test corrections (BH) are used to obtain FDR values (false recovery rates).

Differential expression [37] analysis of two conditions/groups (six biological replicates per condition) was performed using the DESeq2Rpackage (1.20.0). The resulting *p*-values were adjusted using the Benjamini and Hochberg’s approach for controlling the false discovery rate. Genes with an adjusted *p*-value ≤ 0.05 found by DESeq2 were assigned as differentially expressed. The differential gene screening threshold used in this study is |log2(FoldChange)| ≥ 1 & padj ≤ 0.05.

Volcano plots were used to infer the overall distribution of differentially expressed genes. For experiments with biological replicates, the threshold is normally set as padj < 0.05. Through the enrichment analysis of the differentially expressed genes, one can determine which biological functions or pathways are significantly associated with differentially expressed genes. Gene Ontology (GO, http://www.geneontology.org/, accessed on 5 May 2025) is a major bioinformatics initiative to unify the presentation of gene and gene product attributes across all species. DEGs refer to differentially expressed genes. GO enrichment analysis was used by Goseq, which is based on Wallenius’ non-central hypergeometric distribution; its characteristics are as follows: the probability of drawing an individual from a certain category is different from that of drawing it from outside of the category, and this difference can be determined by estimating the preference of gene length.

#### 2.5.8. Enrichment Analysis of Differentially Expressed Genes

Gene Ontology [38] (GO) enrichment analysis of differentially expressed genes was implemented by the clusterProfiler R package version 0.9.1-10, in which gene length bias was corrected. GO terms with corrected Pvalue less than 0.05 were considered significantly enriched by differential expressed genes. KEGG is a database resource for understanding high-level functions and utilities of the biological system, such as the cell, the organism and the ecosystem, from molecular-level information, especially large-scale molecular datasets generated by genome sequencing and other high-throughput experimental technologies (http://www.genome.jp/kegg/; accessed at 13 April 2025). ClusterProfiler R package was used to test the statistical enrichment of differential expression genes in KEGG [39] pathways. The Reactome database brings together the various reactions and biological pathways of human model species. Reactome pathways with corrected Pvalue less than 0.05 were considered significantly enriched by differential expressed genes. The DO (Disease Ontology) database describes the function of human genes and diseases. DO pathways with corrected Pvalue less than 0.05 were considered significantly enriched by differential expressed genes. The DisGeNET database integrates human disease-related genes. DisGeNET pathways with corrected *p*-value less than 0.05 were considered significantly enriched by differential expressed genes.

ClusterProfiler software version 4.12.6 was used to test the statistical enrichment of differentially expressed genes in the Reactome pathway, the DO pathway, and the DisGeNET pathway.

### 2.6. Untargeted Metabolomics

The untargeted metabolomics workflow is presented in Figure 4.

#### 2.6.1. Sample Preparation

All cell culture medium samples (n = 12) were stored at −80 °C until processing. Samples were thawed on ice, lyophilized, and reconstituted in 500 μL of chilled 80% methanol. Following vortexing and 30 min sonication at 4 °C, samples were incubated at −20 °C for 1 h to enhance protein precipitation. After an additional vortexing step (30 s) and incubation at 4 °C for 15 min, samples were centrifuged at 12,000 rpm for 10 min at 4 °C. A total of 200 μL of supernatant was transferred to autosampler vials. A 5 μL aliquot of DL-o-Chlorophenylalanine (0.14 mg/mL) was added as an internal reference standard. Equal aliquots of all samples were pooled to generate quality control (QC) samples, which were processed identically to the study samples.

#### 2.6.2. Instrumental Analysis (UPLC-MS)

Metabolomic profiling was performed using a Vanquish Flex UPLC system coupled to a Q Exactive Plus Orbitrap MS (Thermo Fisher Scientific). Chromatographic separation was achieved using an ACQUITY UPLC H SS T3 column (Waters Corporation, Milford, MA, USA) (100 × 2.1 mm, 1.8 μm). The mobile phase consisted of solvent A (0.05% formic acid in water) and solvent B (acetonitrile), with the following gradient: 0–1 min: 5% B; 1–12 min: 5–95% B; 12–13.5 min: 95% B; 13.5–13.6 min: 95–5% B; 13.6–16 min: 5% B. The flow rate was 0.3 mL/min; column and autosampler temperatures were maintained at 40 °C and 4 °C, respectively.

Mass spectrometry was conducted in both positive (ESI+) and negative (ESI−) ionization modes. Full-scan MS (*m*/*z* 70–1050) was acquired at 70,000 resolution, followed by data-dependent MS/MS (dd-MS2, TopN = 10, resolution 17,500) using higher-energy collisional dissociation (HCD). Ion source parameters: spray voltage 3.0 kV (ESI+), 3.2 kV (ESI−); capillary temperature 350 °C; heater temperature 300 °C; sheath gas 45 arb, aux gas 15 arb, sweep gas 1 arb; S-Lens RF Level: 30% (ESI+), 60% (ESI−).

#### 2.6.3. Data Processing and Quality Control

Raw LC-MS data were processed using Compound Discoverer 3.0 (Thermo) for peak picking, alignment (retention time tolerance ± 0.2 min), and deconvolution. Features were filtered by presence in ≥80% of samples in at least one group. Normalization was performed to total ion current (TIC). QC samples were injected periodically (every 10 runs) to assess analytical reproducibility. The relative standard deviation (RSD) of features in QC samples was calculated, and >80% of features exhibited RSD < 30%, confirming system stability. PCA of QC samples showed tight clustering, supporting reproducibility.

#### 2.6.4. Statistical and Multivariate Analysis

Raw data were processed and aligned using Compound Discoverer (v3.0, Thermo Fisher Scientific) based on accurate *m*/*z* values and retention times. Features from both ESI positive and negative ionization modes were merged and subsequently imported into SIMCA-P software (v14.1, Umetrics, Umeå, Sweden) for multivariate statistical analysis.

An unsupervised Principal Component Analysis (PCA) was first performed to visualize sample distribution and identify potential outliers. To further explore group differences and identify discriminative features, supervised models—Partial Least Squares Discriminant Analysis (PLS-DA) and Orthogonal PLS-DA (OPLS-DA)—were constructed. Candidate biomarkers were selected based on the combination of Variable Importance in Projection (VIP > 1.5), fold change (FC > 2), and statistical significance (*p* < 0.05, two-tailed *t*-test).

Model performance was assessed using R^2^ (explained variance) and Q^2^ (predictive ability). High R^2^ values indicate a good fit between the model and the data, while Q^2^ values reflect the model’s capacity to predict new observations.

#### 2.6.5. Metabolite Identification and Annotation

Metabolite annotation was performed using Compound Discoverer (v3.0, Thermo Fisher Scientific), integrating multiple identification strategies to ensure high confidence in compound assignment. Feature identification relied on accurate precursor *m*/*z* values (mass error < 5 ppm), isotope patterns, retention times (RTs), and MS/MS fragmentation spectra. Spectral matching was conducted against the mzCloud library, with additional database support from Human Metabolome Database (HMDB) and KEGG for biochemical context and pathway mapping.

Both full MS (MS^1^) and data-dependent MS/MS (MS^2^) spectra were utilized for compound annotation. Fragmentation patterns were compared to high-resolution spectral libraries.

The raw data were acquired and aligned using the Compound Discoverer (3.0, Thermo), based on the *m*/*z* values and the retention time of the ion signals. Ions from both ESI− and ESI+ were merged and imported into the SIMCA-P program (version 14.1) for multivariate analysis. A principal component analysis (PCA) was first used as an unsupervised method for data visualization and outlier identification. Supervised regression modeling was then performed on the dataset via partial least squares discriminant analysis (PLS-DA) or orthogonal partial least squares discriminant analysis (OPLS-DA) to identify the potential biomarkers. The biomarkers were filtered and confirmed by combining the results of the VIP values (VIP > 1.5), *t*-tests (*p* < 0.05), and fold-change values (FC > 2). The quality of the fitting model can be explained by the R2 and Q2 values. R2 displays the variance explained in the model and indicates the quality of the fit. Q2 displays the variance in the data, indicating the model’s predictability.

## 3. Results

### 3.1. Lysolecithin Characterization

In Table 1, all data in terms of LPL characterization of the lysolecithin source (soybean based) are documented. More specifically, the characteristics of the aceton insoluable fraction (41.5%) of the lysolecithin source used are depicted. The remaining fraction (58.5%), not characterized in the present study are free fatty acids and the oils.

### 3.2. The Toxicity Screening of Lysolecithins for Intestinal Cells

Figure 5 shows the results of the colorimetric assay for assessing cells’ metabolic activity (MTT). Both the visual check and the protein content indicated that no disruption of the cell monolayer was caused by any of the tested lysolecithin concentrations; however, the mitochondrial respiration was significantly lower upon 1 and 2% lysolecithin treatment compared to the lower doses. Because the effect of 1% lysolecithin treatment was still limited to an approximately 20% decrease in respiration over 24 h of treatment, one might expect reactivity of the cells when only treated for 5 h in the following experiment. Therefore, a concentration of 0.5% lysolecithin was chosen to be applied in the Transwell setup.

### 3.3. The Effect of Lysolecithins on Intestinal Cell Viability

The TEER before incubation was 432 ± 49 Ω/well, while it was 477 ± 112 Ω/well and 489 ± 64 Ω/well after 5 h of incubation with medium alone or with lysolecithin, respectively. No significant differences were observed when comparing before and after treatment, or between treatments. Therefore, one can conclude that the Caco-2 cell monolayer was not disrupted during the experiment. The viability data presented in Figure 6 reveal that lysolecithin effectively improved cell viability compared to the control, suggesting that gene expression pathways that trigger intestinal cell viability were beneficially affected.

### 3.4. The Effects of Lysolecithins on Intestinal Gene Expression

RNA-Seq analysis was performed on eight samples per treatment. The reads were mapped to the human genome and, following the filtering, 14,654 genes were effectively analyzed for differential expression between groups. Data are currently being deposited at GEO.

In Figure 7, the volcano plot illustrates the overall gene expression differences between lysolecithin and the control group. When comparing the control versus lysolecithin, 274 genes were differentially expressed, of which 163 were upregulated and 111 were downregulated (DESeq2 padj ≤ 0.05 |log2FoldChange| ≥ 0.0).

Table 2 contains a list of the top 10 annotated genes (based on *p*-values) with upregulated expression in cells exposed to either control versus lysolecithin.

Table 3 contains a list of the top 10 annotated genes (based on *p*-values) with downregulated expression in cells exposed to the controls versus lysolecithin.

Cluster analysis (see Figure 8) of differential expression indicates genes with similar expression patterns under various experimental conditions. Genes within the same cluster exhibited identical trends in expression levels under different conditions. There is a clear mapping visible in the control (orange bar) versus the lysolecithin samples (pink bar), which implies that the lysolecithin is able to trigger specific genes in the intestinal tract.

The analysis revealed that the overall genomic expression profiles were substantially increased for the Gene Ontology (GO) terms (Figure 9) involved in amino acid transport activities when comparing the lysolecithin-treated groups versus the controls in terms of intestinal cell gene expression. Also, genes involved in receptor signaling and fatty acid metabolism seem to be influenced by the lysolecithin treatment.

The effects of lysolecithins on intestinal metabolite transport/production

The metabolomics data in negative ionization mode showed that the data in the principal component analysis (PCA) plot exhibit a clear grouping trend between the two groups (Figure 10). This implies that the lysolecithin treatment significantly influences metabolite abundance in the basolateral compartment of the in vitro cell culture system.

In the single-variable analysis, the metabolites with VIP > 1.5, FC > 2.0 and *p*-values < 0.05 were selected as significant compounds. Univariate analysis, including fold-change analysis and *t*-tests, was performed on the volcano plot (Figure 11). In negative ionization mode, 220 metabolites were differentially abundant, of which 155 were upregulated and 65 were downregulated. In Appendix A, all metabolites up- and -downregulated have been documented. 

Table 4 and Table 5 contain lists of the top 10 annotated metabolites in the lysolecithin-treated group and the control group, respectively.

### 3.5. Cluster Analysis

Hierarchical cluster analysis of metabolomics data from the control and lysolecithin groups is shown in Figure 12. Color intensity correlates with the degree of increase (red) and decrease (green) relative to the mean metabolite ratio. Extra information on the meaning.

### 3.6. Correlation Network of Metabolites

Under the limiting condition of *p* < 0.05 in MetaboAnalyst V6.0, there are mainly enriched metabolic pathways including the highlighted metabolites, which provide key information for constructing the biomarker network diagram and dot plot (Figure 13).

The metabolomic data in positive ionization mode show that data in the PCA plot exhibit no clear grouping trend between the two groups (Figure 14).

In the single-variable analysis, the metabolites with VIP > 1.5, FC > 2.0 and *p*-values < 0.05 were selected as significant compounds. Univariate analysis, including fold-change analysis and *t*-tests, was performed on the volcano plot (Figure 15). In positive ionization mode, 238 metabolites were differentially abundant, of which 129 were upregulated and 109 were downregulated. Table 6 contains a list of the metabolites with the highest abundance in the lysolecithin treatment group. Extra information is provided here on the relevance of these data.

### 3.7. Cluster Analysis

A hierarchical clustering analysis of metabolomics data from the control group and the lysolecithin group is shown in Figure 16. The mean values of metabolite contents from biological replicates of the control and the lysolecithin group were used to calculate the metabolite ratio. After log transformation of the data, median-centered ratios were normalized. Hierarchical clustering analysis (HCA) was performed using the complete linkage algorithm of the Cluster 3.0 program (Stanford University), and the results were visualized using Pheatmap 1.0.12 (Raivo Kolde). Metabolite ratios of significant metabolites from two independent experiments were used for the HCA. The color intensity corresponds to the degree of increase (red) or decrease (green) relative to the mean metabolite ratio.

### 3.8. Correlation Network of Metabolites

Under the limiting condition of *p* < 0.05 in MetaboAnalyst, there are mainly enriched metabolic pathways including highlighted metabolites, which provide key information for constructing the biomarker network diagram and dot plot (Figure 17). Extra information is provided here on the relevance of these data.

## 4. Discussion

Functional (lyso-)phospholipids (PL and LPL) derived from soy lysolecithin have been hypothesized to serve as signaling compounds [39] that could also trigger intestinal cell gene expression and nutrient absorption via their so-called nutritional gene-regulating action. LPLs are present in cell membranes in small amounts given their role as intermediate precursors in the biosynthesis of other cellular lipids [40]. On the other hand, LPLs are abundantly present in the extracellular environment, where they bind to protein carriers [39]. In eukaryotic cells, LPLs are regarded as bioactive signaling lipids and act as potent messengers in G-protein-coupled signaling pathways [9]. Furthermore, lysolecithin has been proposed to aid in the delivery of polyunsaturated FAs and choline to other tissues [41].

The lysolecithin used in this study was fully screened in terms of PL and LPL characteristics, and the authors also tried to use an in vitro dosage that is physiologically relevant, so that the data collected could be potentially extrapolated to the in vivo situation. However, one should acknowledge that Caco-2 cells lack immune and microbiome context, and in vivo trials are needed to confirm these mechanisms.

The characterization results revealed that 63.08% of the acyl chains present in the studied lysolecithin source were polyunsaturated, mainly including two essential fatty acids: linoleic acid and linolenic acid.

The initial data from the triple-approach experiment showed that the studied lysolecithin can beneficially influence intestinal cells’ viability, which is a first indication of its gene-expression-regulatory potential. The second experiment confirmed that the most significantly affected pathways were related to amino acid transport and metabolism pathways. This highlights that the PL and LPLs present in the lysolecithin can effectively trigger gene expression pathways. Finally, the third experiment showed that the metabolite/nutrient abundance was much higher in the lysolecithin-exposed group compared to the control group. The sample analyses were performed from the basolateral compartment, which tends to emulate the bloodstream. This seems to confirm the nutritional gene expression-regulatory potential of the studied lysolecithin.

Upon thoroughly studying the mode of action of these PL and LPLs, it has become more evident that the different LPL types present in the lysolecithins—such as LPC (lysophosphatidylcholine), LPI (lysophosphatidylinositol), LPE (lysophosphatidylethanolamine), and LPA (lysophosphatidic acid)—can indeed trigger cell signaling pathways and thereby mediate the effects observed at the intestinal tract level [2]. However, it is important to keep in mind that the mixture tested also contains, besides the LPL, a complex mixture of PL compounds (for example, phosphatidylinositols, sphingomyelin, bacterial phospholipids, etc.) that can have a biological impact on the cells.

In the first experiment, the MTT assay measured cellular metabolic activity as an indicator of cell viability, proliferation, and cytotoxicity. The MTT reagent penetrates the cell membrane as well as the mitochondrial inner membrane of viable cells, presumably due to its positive charge and lipophilic structure [42,43], and is reduced to formazan by metabolically active cells. The data showed that the metabolic activity was higher in lysolecithin-treated cells compared to control cells.

The second experiment indicated that the studied lysolecithin can trigger gene expression at a concentration of 0.5%, as it elicited an upregulation of the expression of 163 genes and downregulation of the expression of 111 genes. Upregulation of gene expression refers to an increase in the expression of a gene, leading to a higher level of gene products being produced. This process can occur in response to various stimuli or conditions, resulting in an amplified level of gene expression or the activation of specific pathways. Whether these RNA transcripts are effectively translated into proteins or biological responses needs to be questioned. When attempting to link gene expression data to cell viability data, it is interesting to see that the upregulation of three genes (*PDK4*, *CREB3L3*, and *PLIN2*) is associated with lipid homeostasis and fatty acid metabolism [44]. Additionally, downregulation of *CYP7A1* was observed, which regulates bile biosynthesis [45].

Interestingly, in a previous study, Brautigan et al. [23] reported that in vivo feeding with lysolecithin changed the expression of 135 different genes, which is consistent with the in vitro observations and validates the relevance of the in vitro model used in the present study. The affected pathways coincide when comparing the two studies, which might indicate the presence of a rather large overlap in genes that show significantly affected expression. Furthermore, in the same study [23], it was shown that the addition of LPC alone triggered gene expression to a lesser extent, implying that the LPC in the LPL fraction is not solely responsible for the changes observed in differentially expressed genes. Based on the list of the top 10 genes in terms of affected expression based on log2FC, it is clear that the G-coupled protein receptor and messaging pathways are included. Hence, this study confirms that the mode of action occurs via G-coupled protein signaling cascades [16,46]. It is known that signals are mostly perceived at the membrane level, and transmembrane events are therefore the most plausible routes for signal generation and transduction [47].

When investigating the pathways related to the differentially expressed genes, it becomes clear that the amino acid transporter and the nutrient metabolite pathways are triggered by the lysolecithin. This is very interesting, as it confirms once more that lysolecithin is more than a biosurfactant. Additionally, the already-available in vivo data align with the in vitro observations. Intriguingly, LPC has the ability to improve intestinal morphology and nutrient digestion and absorption [2,7,48]. Nutautaite et al. [49] documented significant improvements in intestinal villus height. The addition of LPC reduced the crypt depth, increased the jejunal villi’s height, and improved the ratio of villus height to crypt depth in the jejuna and duodena of chickens. Very interestingly, LPC upregulated the expression of amino acids (AAs) and cholesterol transporter genes in enterocytes, as well as increased fat’s digestibility and the intake of cholesterol and amino acids [50]. In addition, Zhang et al. [51] recently showed that LPLs significantly increase the ileal digestibility of AAs, including Ile, Thr, Phe, His, Arg, Tyr, Glu, Pro, Gly, and Ala. Furthermore, the gene expression of amino acid transporters was significantly elevated as a result of LPL supplementation.

It is important to keep in mind, when interpreting the RNA-sequencing data, that the correlation between expression levels of protein and mRNA in mammals is relatively low [52]. Suggested explanations for this low correlation include post-transcriptional regulation and measurement noise. This low correlation makes it difficult to integrate protein and mRNA data [52]. Hence, one needs to be careful when considering the consequences and impact of the intestinal tract transcriptome data presented in the current study. Also, given the commercial involvement in this study, future independent replication is essential to confirm the reported findings.

Detailed results on metabolite abundance in the basolateral compartment of the intestinal cell culture setup exhibit the increased abundance of metabolites and nutrients upon lysolecithin exposure. In particular, the metabolites upregulated by the lysolecithin treatment showed a much higher abundance (reflected in a greater FC) than the downregulated metabolites. This higher abundance can be ascribed to three reasons: (1) increased transport or (2) increased metabolic activity of enterocytes in response to the lysolecithin, or (3) increased absorption surface area. A subsequent step could consist of an investigation of the bioavailability of nutrients in an in vivo trial. The fact that the lysolecithin seems to beneficially affect the intestinal morphology could also explain the increased uptake of nutrients throughout the intestinal tract [53]. However, the metabolite data, in combination with the gene expression data, suggest that amino acid transport was also significantly increased in the lysolecithin-treated intestinal cells. However, when looking more in depth at Figure 9, Figure 13 and Figure 17, it becomes more and more clear that in the intestinal cells, amino acid transporters and genes related to lipid metabolism are increasingly expressed in the lysolecithin treatment versus controls. By contrast, in the metabolite profile pathways, one can see more prominently the increased lipid metabolite pathways being affected. Interestingly, when digging into the metabolite abundance data, it seems that specific LPL fractions (such as LysoPC(18:3) and LysoPA(20:2) are more abundant in the LL group compared to the control group. This might explain the effect of LL not only in the intestinal tract but also beyond (liver, immune system, ...) [54,55].

It remains unclear which of the PL or LPL fractions induces the action observed in the present study. The lysolecithin used in this study was fully characterized in terms of PL and LPL composition and FA distribution, using high-performance liquid chromatography (HPLC), gas chromatography (GC), and thin-layer chromatography (TLC). However, there is a lack of available literature on the ideal composition and/or ratios of the LPLs required to maximize their nutritional gene-regulatory impact on animals. This is expected to be the subject of further studies.

An increasing number of studies highlight the importance of the quality of the PL and LPLs, which is highly dependent on their biochemical structure, rather than their level of inclusion in the diet. For example, lysophosphatidylcholine (LPC) may have inflammatory properties, which appear to be highly dependent on its FA composition [9]. As reviewed by Drzazga et al. [24], pro-inflammatory properties have been mainly associated with LPC containing saturated and monounsaturated FAs, especially C14:0 and C16:0. Remarkably, anti-inflammatory properties have been observed with LPC containing polyunsaturated FAs, especially 22:4 and 22:6, which have been reported to neutralize inflammatory effects induced by saturated LPC C16:0 in vivo [9,25]. Lysophosphatidylethanolamine (LPE), a monounsaturated LPL (C18:1), has been shown to express similar inflammatory properties as those found for LPC [25].

Moreover, there is an indication that the gene expression-regulatory potential of lysolecithins may exceed the intestinal tract. Zhang et al. [51] recently showed that LPLs significantly increased the gene expression of growth hormone and insulin-like growth factor 1. The latter could be a crucial additional effect of lysolecithins, although more research is necessary to further explore this mode of action. It is also suggested that metabolites originating from polyunsaturated lysolecithins may be responsible for their anti-inflammatory properties. On the other hand, saturated lysolecithins may exert pro-inflammatory action, primarily due to their effects on endothelial cells, leading to a reversible opening of the cells’ tight junctions and thereby to plasma leakage and leukocyte migration. The expression of adhesion molecules and the release of chemotactic factors are other pro-inflammatory actions associated with saturated and monounsaturated LPC [9,24]

In conclusion, these data suggest that the impact of lysolecithin on intestinal tract cells cannot solely be ascribed to its surface chemistry action. The data presented in this study reveal, for the first time, how PL and LPLs present in lysolecithin can trigger gene expression and intestinal cell function. The gene expression data revealed that the nutrient transporters were upregulated in the lysolecithin-treated intestinal cells. This could potentially facilitate the increased transport of metabolites from the apical lumen (which reflects the intestinal lumen) towards the basolateral compartment (which reflects the bloodstream). Overall, this study seems to reveal that lysolecithin is a mixture of bioactive compounds that can trigger gene expression and metabolism in the intestinal tract. However, extra in vivo data are needed to corroborate the in vitro findings.

## Figures and Tables

**Figure 1 cimb-47-00548-f001:**
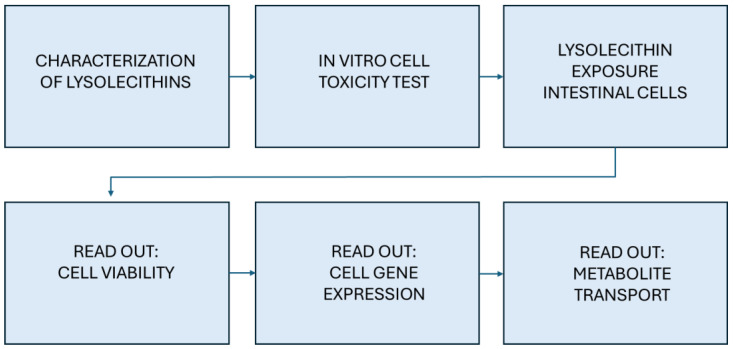
Block diagram of the experimental flow executed for this study.

**Figure 2 cimb-47-00548-f002:**
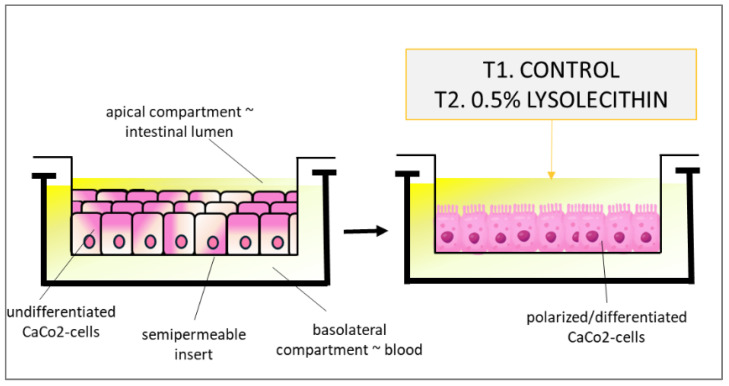
Experimental setup of the polarized cell culture system.

**Figure 3 cimb-47-00548-f003:**
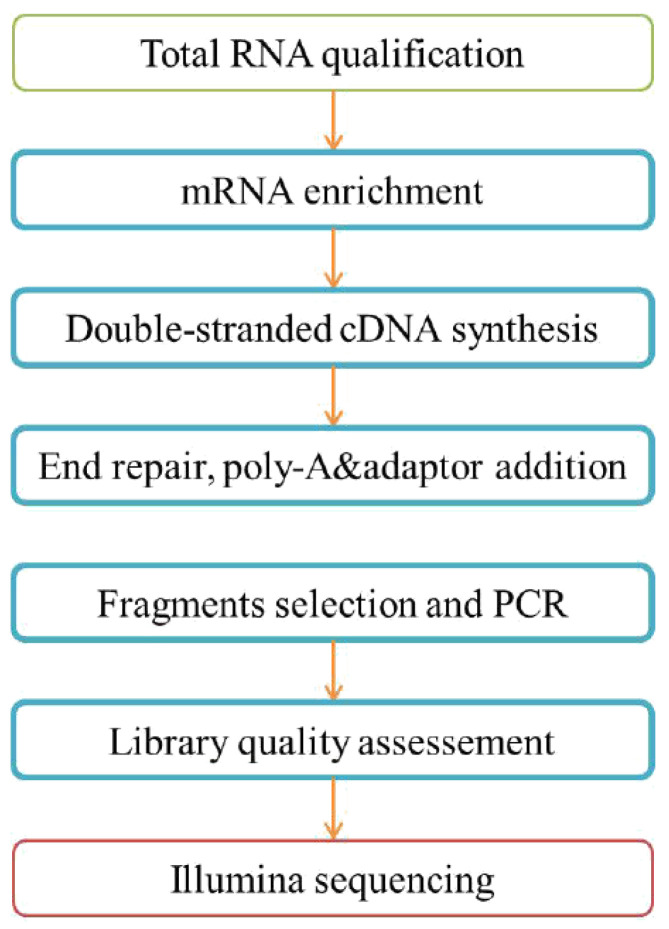
Workflow of the RNA sequencing.

**Figure 4 cimb-47-00548-f004:**
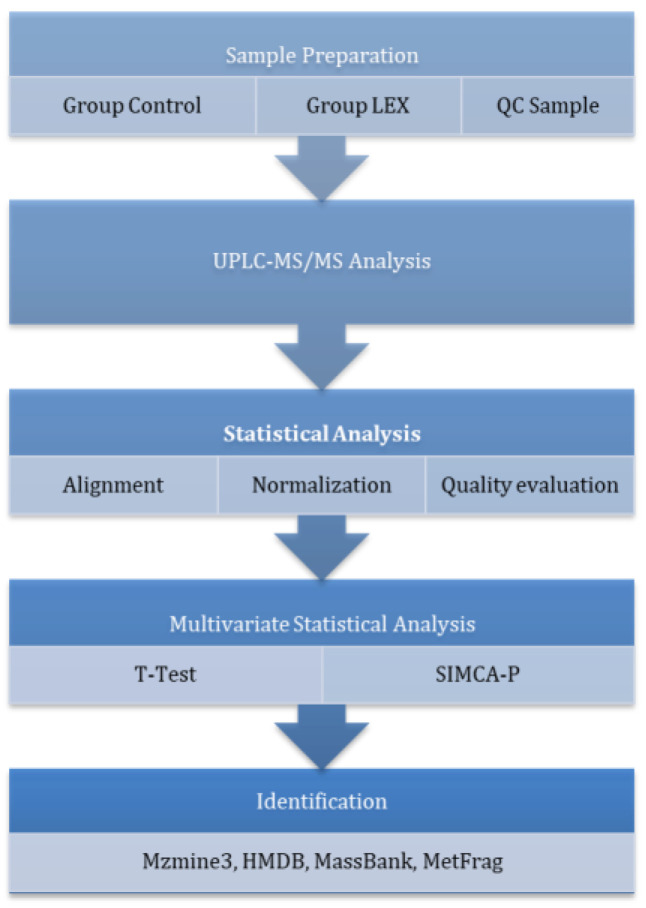
Workflow for the untargeted proteomics analyses.

**Figure 5 cimb-47-00548-f005:**
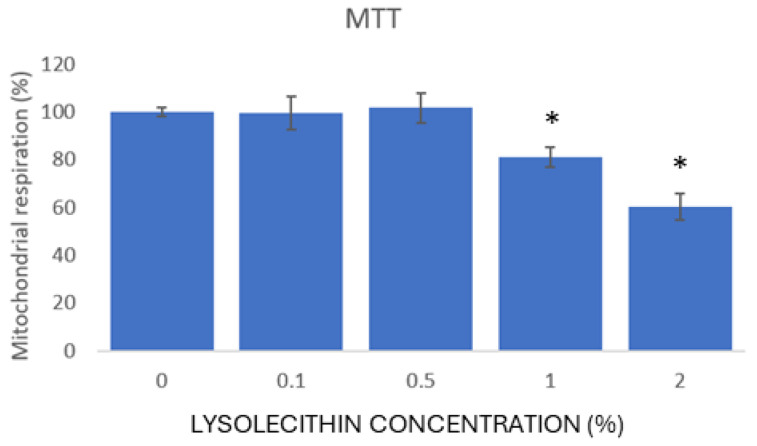
Toxicity assay data based on MTT absorbance. MTT absorbance values relative to the untreated condition; n = 12; * shows values significantly different (*p* < 0.05) from the untreated condition according to ANOVA. Post-hoc comparisons between treatments were carried out using Duncan’s test.

**Figure 6 cimb-47-00548-f006:**
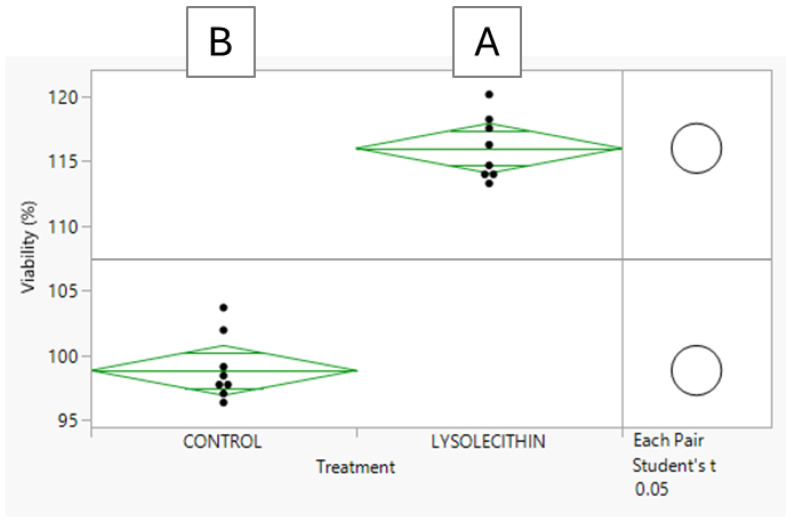
Intestinal cell viability of Caco-2 cells in response to 0.5% lysolecithin versus control; n = 12; A and B values show significant differences between treatments (*p* < 0.05) according to Student’s *t*-test.

**Figure 7 cimb-47-00548-f007:**
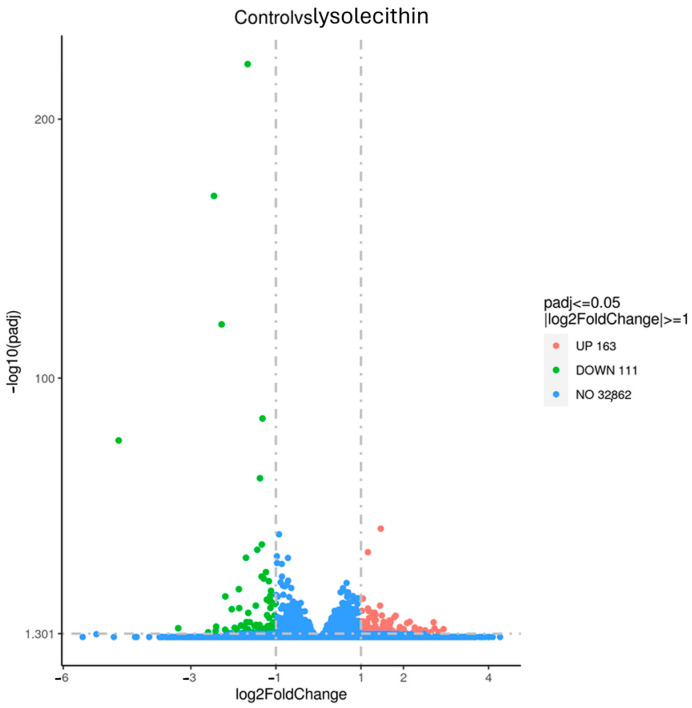
Volcano plot for comparison of control versus lysolecithin (n = 12). Padjusted ≤ 0.05 and log2FC ≥ 1. *Y*-axis: It is constructed by plotting the negative logarithm (base 10) of the *p*-value on the *y*-axis, ensuring that data points with lower *p*-values—indicative of higher statistical significance—are positioned toward the top of the plot. The *x*-axis of the volcano plot represents the fold change in gene expression between two conditions. Fold change quantifies how much the expression of a gene has increased or decreased.

**Figure 8 cimb-47-00548-f008:**
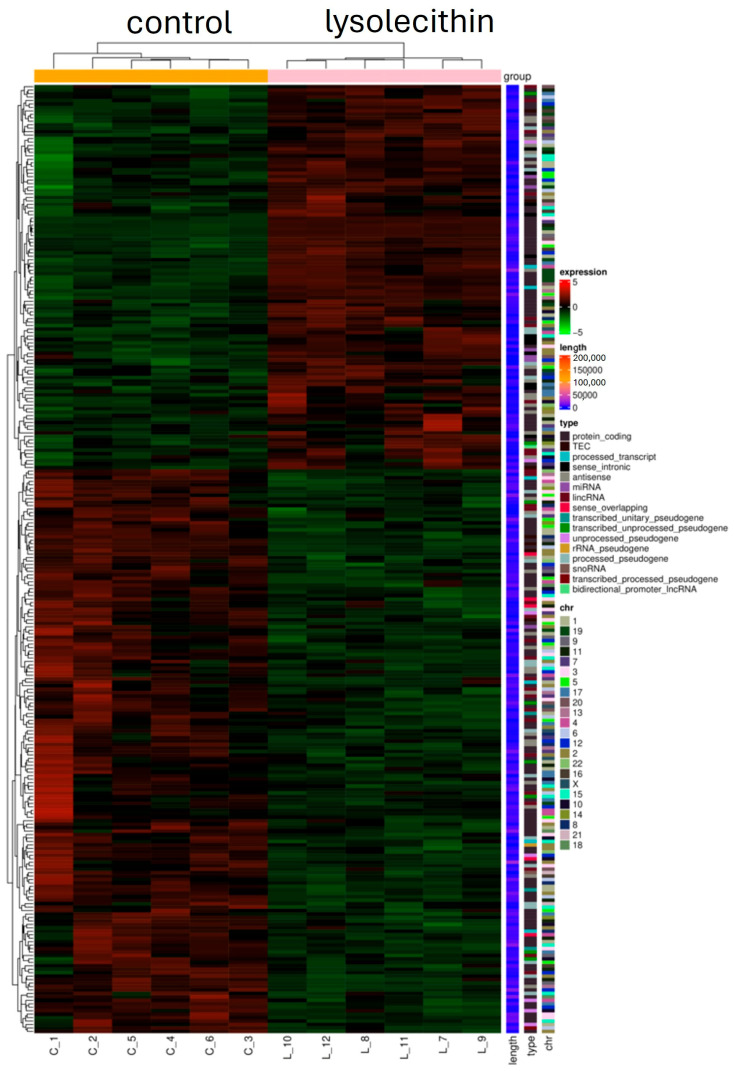
Heat cluster: C_ samples are all control samples (orange bar). L_ samples are the lysolecithin samples (pink bar). N = 12. Genes are clustered according to their similarity. The color indicates a change from the mean value: red is an increase, green is a decrease.

**Figure 9 cimb-47-00548-f009:**
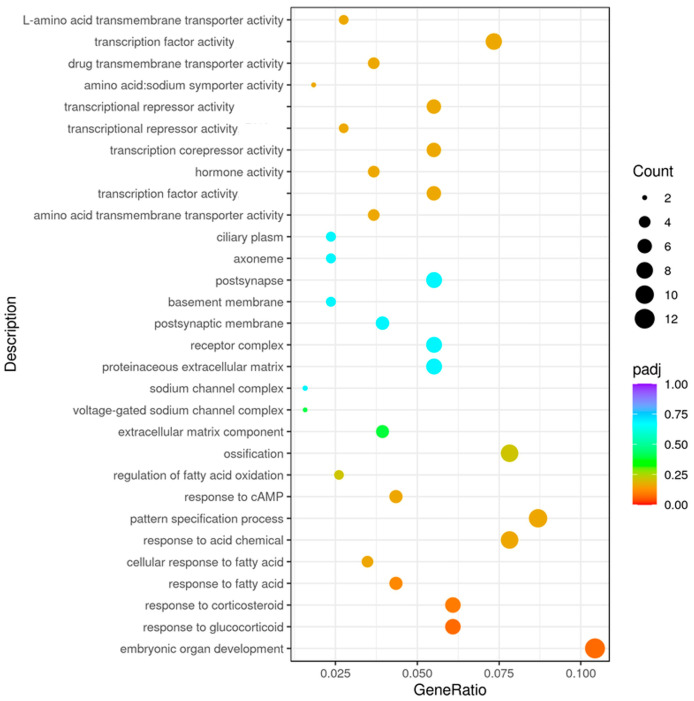
Enrichment analysis with GO terms comparing the lysolecithin-treated group versus the control. Enriched GO terms: dot plot. The 25 GO processes with the largest gene ratios are plotted in order of gene ratio. The size of the dots represent the number of genes in the significant DE gene list associated with the GO term and the color of the dots represent the P-adjusted values (BH).

**Figure 10 cimb-47-00548-f010:**
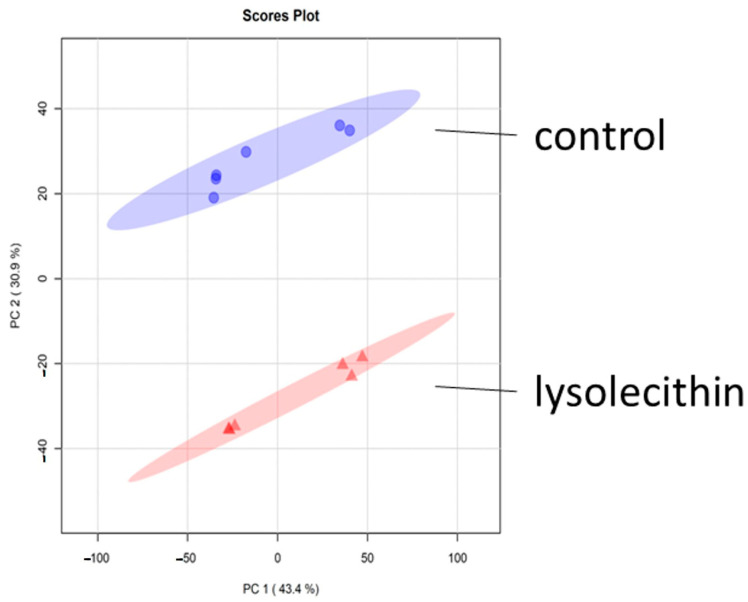
The score scatterplot of the PCA model, comparing the lysolecithin-treated group versus the control. All pairwise sample Pearson correlations are shown in the X- and the *Y*-axis.

**Figure 11 cimb-47-00548-f011:**
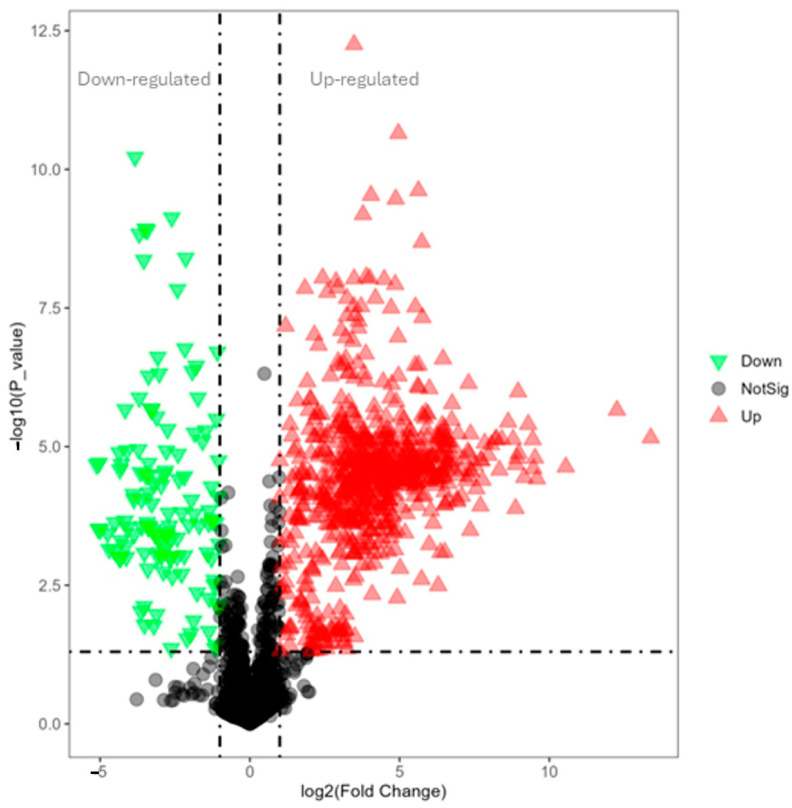
Volcano plot for the lysolecithin group versus the control: Red represents the upregulated metabolites in the lysolecithin group compared to the control, green represents the downregulated metabolites in the lysolecithin group compared with the control, and gray represents the metabolites with no difference between the control group and the lysolecithin group. Treshold with VIP > 1.5, FC > 2.0 and *p*-values < 0.05. *Y*-axis: It is constructed by plotting the negative logarithm (base 10) of the *p*-value on the *y*-axis, ensuring that data points with lower *p*-values—indicative of higher statistical significance—are positioned toward the top of the plot. The *x*-axis of the volcano plot represents the fold change in gene expression between two conditions. Fold change quantifies how much the expression of a gene has increased or decreased. The dashed lines indicate the treshold for up- and downregulated metabolites.

**Figure 12 cimb-47-00548-f012:**
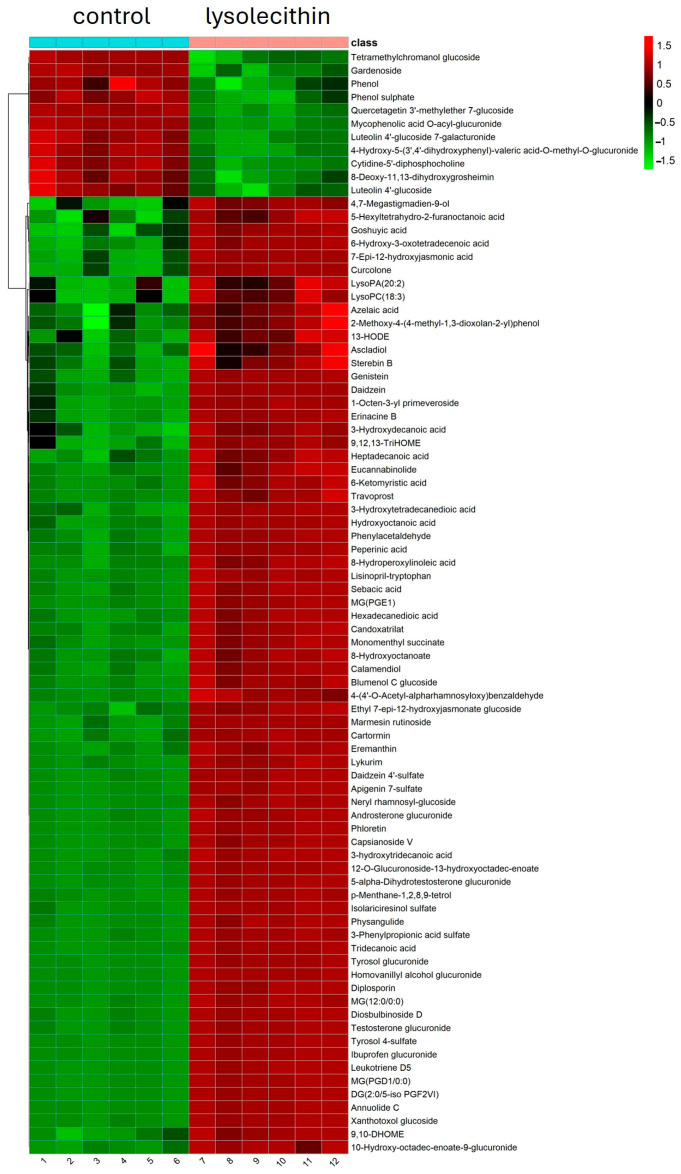
Hierarchical cluster analysis of metabolome data from significant metabolites comparing the lysolecithin-treated (pink bar) group versus the control (blue bar). Metabolites are clustered according to the similarity. The color indicates a change from the mean value. Red is an increase, green is a decrease.

**Figure 13 cimb-47-00548-f013:**
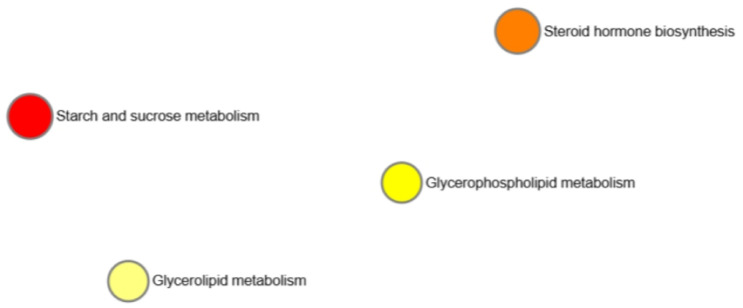
Network diagram of enrichment analysis in metabolites. The color depicts how drastically metabolites of a certain pathway are over-represented (the darker, the bigger the effect).

**Figure 14 cimb-47-00548-f014:**
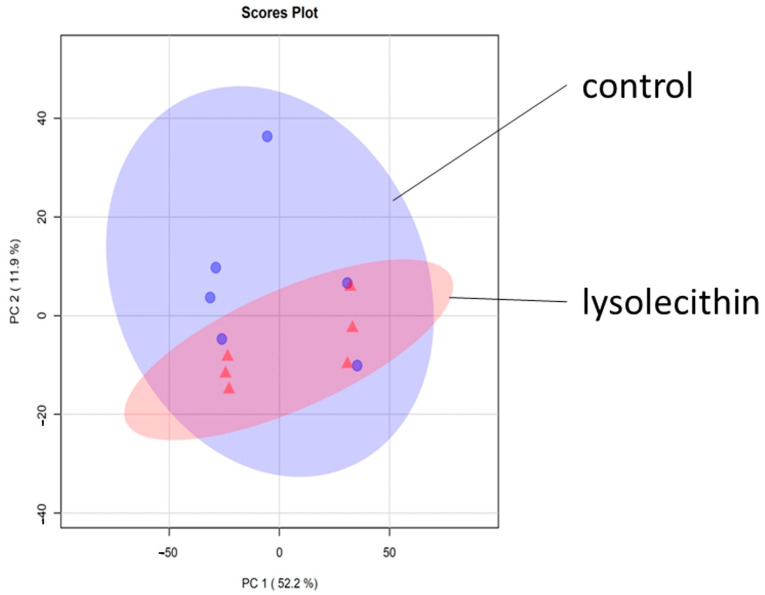
The score scatterplot of the PCA model comparing the lysolecithin-treated group versus the control. All pairwise sample Pearson correlations are shown in the X- and the *Y*-axis. The blue circles depict the control samples and the red triangles depict the lysolecithin samples.

**Figure 15 cimb-47-00548-f015:**
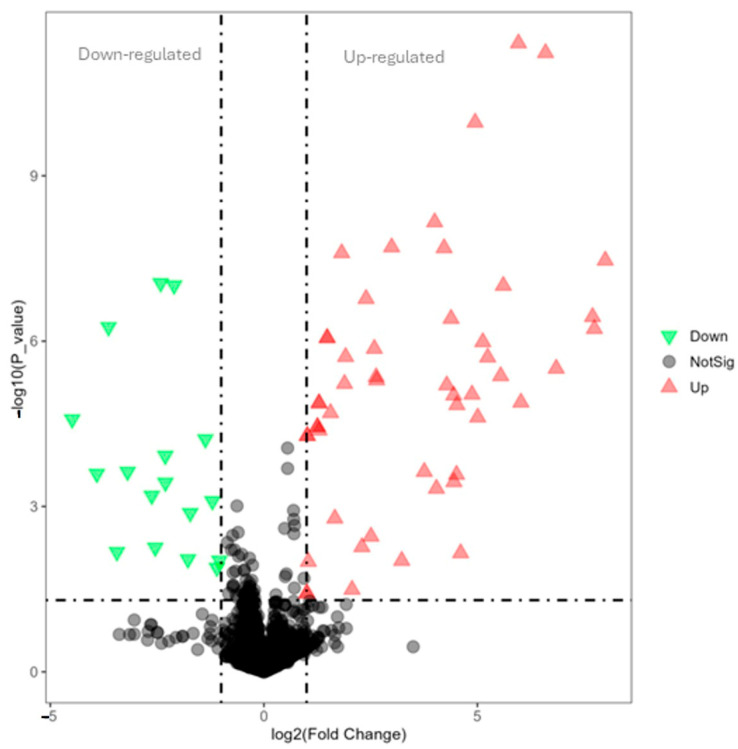
Volcano plot for the lysolecithin group versus the control. Red represents the upregulated metabolites in the lysolecithin group compared to the control, green represents the downregulated metabolites in the lysolecithin group compared to the control, and gray represents the metabolites with no difference between the control group and the lysolecithin group. Treshold with VIP > 1.5, FC > 2.0 and *p*-values < 0.05. The dashed lines indicate the treshold for up- and downregulated metabolites.

**Figure 16 cimb-47-00548-f016:**
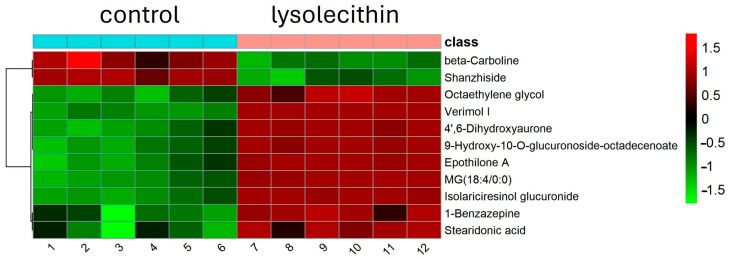
Hierarchical cluster analysis of metabolomics data from significant metabolites. Metabolites are clustered according to their similarity. The color indicates a change from the mean value. Red is an increase, while green is a decrease.

**Figure 17 cimb-47-00548-f017:**
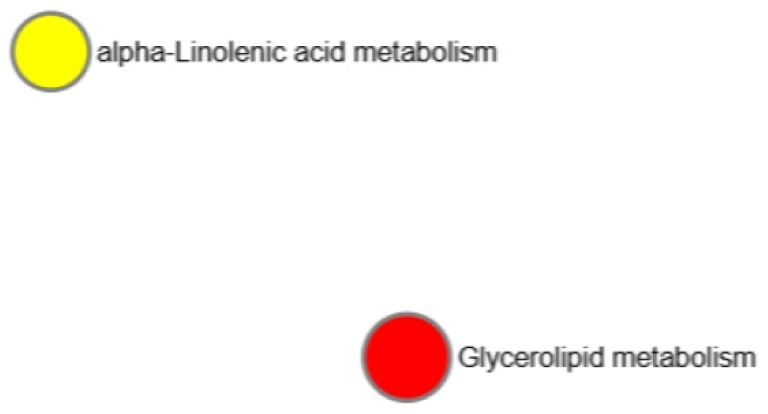
Network diagram of enrichment analysis in metabolites. The color depicts how drastically metabolites of a certain pathway are over-represented (the darker, the bigger the effect).

**Table 1 cimb-47-00548-t001:** Lysolecithin composition based on LPL profile and LPL fatty acid chains.

Phospholipid Profile of the Lysolecithin (%)		Fatty Acid Chains on Lysophospholipids (%)	
Phosphatidylglycerols	0.72	14:0	0.07
Phosphatidylethanolamines	8.84	15:0	<0.05
Phosphatidylinositols	7.23	16:0	15.57
Phosphatidylserins	<1	16:1	0.14
Lipid A phospholipids	1.67	17:0	0.14
Phosphatidylcholines	12.06	17:1	0.05
Sphingomyelins	<0.4	18:0	4.22
Lysophosphatidylcholines	5.08	18:1 trans	<0.05
Lysophosphatidylethanolamines	3.75	18:1	15.52
Lysophosphatidylinositols	2.19	18:2 trans	0.09
Total	41.54	18:2 (n − 6)	56.15
		18:3 trans	<0.05
		18:3 (n − 3)	6.93
		20:0	0.24
		20:1	0.15
		20:2 (n − 6)	<0.05
		22:0	0.40
		22:1	<0.05
		24:0	0.22
		24:1	<0.05
		Content (g/100 g)	68.52
		Saturated fatty acids (%)	20.87
		Monounsaturated fatty acids (%)	15.86
		Polyunsaturated fatty acids (%)	63.08
		Of which (n − 6) (%)	56.15
		Of which (n − 3) (%)	6.93

**Table 2 cimb-47-00548-t002:** List displaying the top 10 annotated genes based (to which a function could be assigned) on log2FC value (Padj < 0.05) with upregulated expression in lysolecithin-treated intestinal cells.

Gene_ID	log2FoldChange	padj	Gene_Name	Gene_Description
ENSG00000004799	−4.70	0.0000	*PDK4*	pyruvate dehydrogenase kinase 4 [Source: HGNC Symbol; Acc:HGNC:8812]
ENSG00000143333	−3.30	0.0004	*RGS16*	regulator of G protein signaling 16 [Source: HGNC Symbol;Acc:HGNC:9997]
ENSG00000060566	−2.46	0.0000	*CREB3L3*	cAMP responsive element binding protein 3 like 3 [Source: HGNC Symbol; Acc:HGNC: 18855]
ENSG00000279357	−2.42	0.0049	*AC007224.2*	TEC
ENSG00000199047	−2.41	0.0001	*MIR378A*	microRNA 378a [Source: HGNC Symbol;Acc:HGNC:31871]
ENSG00000147872	−2.28	0.0000	*PLIN2*	perilipin 2 [Source: HGNC Symbol;Acc:HGNC:248]
ENSG00000259771	−2.19	0.0016	*AC092756.1*	novel transcript, sense intronic to MYO1E
ENSG00000268635	−2.07	0.0355	*AP003680.1*	novel transcript, antisense to PAK1
ENSG00000103044	−2.04	0.0000	*HAS3*	hyaluronan synthase 3 [Source: HGNC Symbol; Acc:HGNC:4820]
ENSG00000168505	−1.97	0.0003	*GBX2*	gastrulation brain homeobox 2 [Source: HGNC Symbol; Acc:HGNC:4186]

**Table 3 cimb-47-00548-t003:** List displaying the top 10 annotated genes (to which a function could be assigned) based on log2FC value (Padj < 0.05) with downregulated expression in lysolecithin-treated intestinal cells.

Gene_ID	log2FoldChange	padj	Gene_Name	Gene_Description
ENSG00000136546	2.94	0.0009	*SCN7A*	sodium voltage-gated channel alpha subunit 7 [Source:HGNC Symbol;Acc:HGNC:10594]
ENSG00000109743	2.82	0.0108	*BST1*	bone marrow stromal cell antigen 1 [Source:HGNC Symbol;Acc:HGNC:1118]
ENSG00000167910	2.73	0.0005	*CYP7A1*	cytochrome P450 family 7 subfamily A member 1 [Source:HGNC Symbol;Acc:HGNC:2651]
ENSG00000256651	2.71	0.0000	*AC006518.1*	taste receptor, type 2 pseudogene
ENSG00000081148	2.54	0.0206	*IMPG2*	interphotoreceptor matrix proteoglycan 2 [Source:HGNC Symbol;Acc:HGNC:18362]
ENSG00000154263	2.51	0.0026	*ABCA10*	ATP binding cassette subfamily A member 10 [Source:HGNC Symbol;Acc:HGNC:30]
ENSG00000235079	2.47	0.0117	*ZRANB2-AS1*	ZRANB2 antisense RNA 1 [Source:HGNC Symbol;Acc:HGNC:43594]
ENSG00000198939	2.39	0.0011	*ZFP2*	ZFP2 zinc finger protein [Source:HGNC Symbol;Acc:HGNC:26138]
ENSG00000136531	2.32	0.0235	*SCN2A*	sodium voltage-gated channel alpha subunit 2 [Source:HGNC Symbol;Acc:HGNC:10588]
ENSG00000168542	2.27	0.0318	*COL3A1*	collagen type III alpha 1 chain [Source:HGNC Symbol;Acc:HGNC:2201]

Note: Uncharacterized Transcripts in Table 2 and Table 3 were removed so that focus relies on biologically interpretable genes.

**Table 4 cimb-47-00548-t004:** List of the annotated metabolites based on Foldchange threshold 2–20x (Padj < 0.05) with abundance following lysolecithin treatment in negative ionization mode.

HMDB_ID	Compound_Name	Chemical Formula	FC (Lysolecithin/Control)	Log2(FC)	*t*-Test
HMDB0000792	Sebacic acid	C_10_H_18_O_4_	19.31	4.27	0.0001
HMDB0249582	Candoxatrilat	C_20_H_33_NO_7_	16.90	4.08	0.0002
HMDB0004704	9,10-DHOME	C_18_H_34_O_4_	16.17	4.02	0.0002
HMDB0000672	Hexadecanedioic acid	C_16_H_30_O_4_	15.05	3.91	0.0002
HMDB0004708	9,12,13-TriHOME	C_18_H_34_O_5_	14.64	3.87	0
HMDB0302703	Eremanthin	C_15_H_18_O_2_	13.83	3.79	0.0001
HMDB0303565	Eucannabinolide	C_22_H_28_O_8_	13.81	3.79	0.0016
HMDB0000560	Goshuyic acid	C_14_H_24_O_2_	12.26	3.62	0
HMDB0062363	6-Hydroxy-3-oxotetradecenoic acid	C_14_H_24_O_4_	11.48	3.52	0
HMDB0006236	Phenylacetaldehyde	C_8_H_8_O	10.78	3.43	0.0001
HMDB0034673	Calamendiol	C_15_H_26_O_2_	9.07	3.18	0.0001
HMDB0004706	8-Hydroperoxylinoleic acid	C_18_H_32_O_4_	8.68	3.12	0.0002
HMDB0010387	LysoPC(18:3)	C_26_H_48_NO_7_P	8.08	3.01	0.0242
HMDB0040668	Blumenol C glucoside	C_19_H_32_O_7_	7.30	2.87	0.0002
HMDB0114758	LysoPA(20:2)	C_23_H_43_O_7_P	7.24	2.86	0.049
HMDB0036143	Monomenthyl succinate	C_14_H_24_O_4_	5.66	2.50	0.0001
HMDB0002259	Heptadecanoic acid	C_17_H_34_O_2_	5.44	2.44	0.0003
HMDB0030982	6-Ketomyristic acid	C_14_H_26_O_3_	4.77	2.25	0.0003
HMDB0038731	4,7-Megastigmadien-9-ol	C_13_H_22_O	4.67	2.22	0.0001
HMDB0010725	3-Hydroxydecanoic acid	C_10_H_20_O_3_	4.05	2.02	0
HMDB0004667	13-HODE	C_18_H_32_O_3_	3.71	1.89	0.0027
HMDB0061914	8-Hydroxyoctanoate	C_8_H_16_O_3_	3.58	1.84	0
HMDB0031127	5-Hexyltetrahydro-2-furanoctanoic acid	C_18_H_34_O_3_	3.33	1.73	0.0004
HMDB0036199	2-Methoxy-4-(4-methyl-1,3-dioxolan-2-yl)phenol	C_11_H_14_O_4_	3.31	1.73	0.0008
HMDB0000784	Azelaic acid	C_9_H_16_O_4_	3.08	1.62	0.0008
HMDB0035338	Sterebin B	C_20_H_32_O_5_	2.33	1.22	0.0015
HMDB0029610	Ascladiol	C_7_H_8_O_4_	2.02	1.01	0.0112

**Table 5 cimb-47-00548-t005:** List of the annotated metabolites based on Foldchange threshold 2–20x (Padj < 0.05) with decreased abundance after lysolecithin treatment in negative ionization mode.

HMDB_ID	Compound_Name	Chemical Formula	FC (Contol/Lysolecithin)	Log2(FC)	*t*-Test
HMDB0059972	4-Hydroxy-5-(3′,4′-dihydroxyphenyl)-valeric acid-O-methyl-O-glucuronide	C_18_H_26_O_12_	17.96	4.51	0.0025
HMDB0060491	Mycophenolic acid O-acyl-glucuronide	C_23_H_28_O_12_	17.31	4.36	0.0001
HMDB0260340	Cytidine-5′-diphosphocholine	C_14_H_26_N_4_O_11_P_2_	15.80	4.32	0.0033
HMDB0038866	Quercetagetin 3′-methylether 7-glucoside	C_22_H_22_O_13_	10.17	3.82	0.0000
HMDB0258916	Tetramethylchromanol glucoside	C_20_H_30_O_7_	9.86	3.49	0.0000
HMDB0252642	Gardenoside	C_17_H_24_O_11_	7.49	3.42	0.0000
HMDB0038809	Luteolin 4′-glucoside 7-galacturonide	C_27_H_28_O_17_	4.18	2.18	0.0002

**Table 6 cimb-47-00548-t006:** List displaying the top annotated metabolites. Foldchange threshold 2–20x (Padj < 0.05) with increased abundance after lysolecithin treatment in positive ionization mode.

HMDB_ID	Compound_Name	Chemical Formula	FC(LEX/Control)	Log2(FC)	*t*-Test
HMDB0094680	Octaethylene glycol	C_16_H_34_O_9_	6.23	2.64	0.0005
HMDB0244865	1-Benzazepine	C_10_H_9_N	3.70	1.89	0.0006
HMDB0006547	Stearidonic acid	C_18_H_28_O_2_	2.95	1.56	0.0019

## Data Availability

Data is contained within the article or Appendix A.

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
