# Peer review of "The Nutritional Gene Expression Regulation Potential of a Lysolecithin-Based Product"

_cimb, 2025, doi:10.3390/cimb47070548_

Round 1
Reviewer 1 Report
Comments and Suggestions for Authors
1.Abstract.
(1)“the potential of a lysolecithin .... in the intestinal tract”. This is an inaccurate expression. Do you mean the regulation of intestinal health? Intestinal immunity? It should be written accurately.
(2)“Lysolecithin is a performance-enhancing product for livestock”. This sentence should be placed before "The scope of this study".
2.Introduction:
(1)Research should be based on a hypothesis.
(2)The purpose and significance of this study should be described more clearly.
3.Materials and Methods
(1)The information such as the source and purity of lysolecithin should be provided in detail.
(2)Is there any research basis for the dosage setting of lysolecithin?
(3)The methods of transcriptome and metabolomics should be better written in the form of articles, not an experimental process.
(4)Statistical method: All groups should do statistical analysis and multiple comparisons together, instead of just comparing the treatment group and the control group.
4.The results are not well presented. The differences of genes and metabolites between the treatment group and the control group, especially the pathways and representative biomarkers, should be demonstrated. Instead of describing some methods or explaining some analysis, this should not belong to the result section. This part should be rewritten.
The English could be improved to more clearly express the research.
Author Response
Dear reviewer,
We genuinely want to thank you for your thorough revision of our manuscript. All of the comments that you have provided are very relevant, and we truly believe that they helped to increase the quality of the paper and reach the level needed for publication.
First of all, the authors started working on the general comments provided on the manuscript. Main efforts carried out the general text are:
- The authors improved the quality of English language use through external revision using the MDPI English editing service.
- The entire manuscript has been rewritten to make sure that no suggestive sentences were included in the revised manuscript. Furthermore, additional recent references have been added to the manuscript, such that it can be considered more up-to-date.
- The research design has been better described in the reviewed paper, thus increasing its quality.
- The Methods have been completely rewritten based on the requests from the reviewer to improve reproducibility. Extra processes have been included, allowing the reader to better replicate the experiments from scratch.
- The Results have been presented in an amended way, making them clearer to the readers. Also results are more into depth discussed in the discussion section. All method related parts have been removed from the Results section.
- The Conclusions have been better linked to the results. And also, conclusions withdrawn from the in vitro dataset have been softened and it has been clearly stated that there are limitation with regards to interpretation of RNAseq data and also towards extrapolation to the in vivo situation.
Regarding the more specific comments received, we have made the needed changes and provided the reviewers with the associated responses in the following.
With regards to the abstract:
#(1) “the potential of a lysolecithin .... in the intestinal tract”. This is an inaccurate expression. Do you mean the regulation of intestinal health? Intestinal immunity? It should be written accurately.
→ The authors do fully agree with this comment and have rephrased this sentence into: The scope of this study was to fully understand the potential of a lysolecithin source derived from soybeans (present in LYSOFORTE®) to influence intestinal nutrient transport in the intestinal tract.
#(2)“Lysolecithin is a performance-enhancing product for livestock”. This sentence should be placed before "The scope of this study".
→ Indeed, the authors have transferred this sentence to the beginning of the abstract.
Then, for the Introduction:
#(1)Research should be based on a hypothesis.
→ An extra sentence has been added to the Introduction to rephrase the real hypothesis of the presented work. “The hypothesis of this study is that lysolecithins can have bioactive potential on intestinal cells by which nutrient metabolism and transport are being regulated.”
#(2)The purpose and significance of this study should be described more clearly.
→ The authors agree and have added an extra section to end the Introduction part: “Hereby, it was clarified that the effects of the lysolecithins exceed surface chemistry; as such, lysolecithins contain compounds that seem to have very potent signaling functions. This would imply that very small amounts of the right lysolecithins can have a significant impact on the intestinal tract metabolism and nutrient transport actions.”
With regards to the Materials and Methods:
#(1)The information such as the source and purity of lysolecithin should be provided in detail.
→ The source of lysolecithin is soybean. The soybean-lecithin has been treated with phospholipase A2 to generate lysolecithin. This has additionally been specified in the manuscript. The 41.5% value is exactly the aceton insoluble matter of a typical lysolecithin product, which means consisting of the phospholipids, glycolipids, and carbohydrates all together. The rest of the lysolecithin is oil and fatty acids which dissolve in acetone (references available at bottom). The latter fraction is not depicted in the Table as this is not relevant for the bioactivity. This has been addressed in the results section of the paper.
#(2)Is there any research basis for the dosage setting of lysolecithin?
→ The authors included an extra section on the relevance of the dose setting in this study in the methods section. “The in vivo addition of LYSOFORTE in broiler diets is 250g up to 500g/t. A young broiler chicken eats 200g of feed per day. This implies that the broiler will have an intake of ±0.5g lysolecithin per day. The content of intestinal lumen in the broiler is estimated to be 100ml. So that means 0.5g/100ml = 0.5% exposure of lysolecithin in the intestinal tract per day. Hence, the dosage set up in this in vitro work is physiologically relevant for the in vivo situation.”
#(3)The methods of transcriptome and metabolomics should be better written in the form of articles, not an experimental process.
→ The experimental procedures for transcriptomics and metabolomics have been completely revised and rewritten so that this valid remark should be solved. All revisions are indicated in blue in the manuscript.
When looking at the Results:
#(1)The results are not well presented. The differences of genes and metabolites between the treatment group and the control group, especially the pathways and representative biomarkers, should be demonstrated. Instead of describing some methods or explaining some analysis, this should not belong to the result section. This part should be rewritten.
→ This is again a very valid remark. Hence, to tackle this issue, the entire Results section has been rewritten. Also, all pathways and biomarkers have been reintegrated into the Discussion section where they are adequately discussed. All method related parts have been removed from the Results section as much as possible. Also for the gene expression data, Uncharacterized Transcripts in Tables 2 and 3: were removed so that focus relies on biologically interpretable genes. The latter have then consequently been discussed in the Discussion section.
→ Furthermore, the researchers spent extra time in improving the methods and results sections as requested in the general overview:
- For the Lysolecithin characterization work: Extra data on the analyses have been provided so that it will be easier to reproduce the analyses.
- For the RNAseq: The methods section and results section for this subjects have been rewritten so that all reviewers concerns have been addressed as much as possible.
- For the metabolomic work: The Methods section has been extended so that extra information on the analysis of the markers have been provided. Also, abnormally highly abundant metabolites have been removed with a cutoff level so that only significant data that are also relevant have been taken into account.
- For the Results section: This part has been completely rewritten based on the reviewer’s suggestions. The authors do believe that the quality of the paper is improved drastically.

Reviewer 2 Report
Comments and Suggestions for Authors
Thank you for submitting your manuscript entitled “The Nutritional Gene Regulation Potential of a Lysolecithin-Based Product.” The study explores an important and under-investigated aspect of lysolecithins—specifically their potential role in regulating intestinal gene expression and nutrient metabolism beyond surface emulsification. By employing a combination of in vitro assays using Caco-2 cells, transcriptomic profiling, and untargeted metabolomics, the authors attempt to demonstrate the bioactivity of a soybean-derived lysolecithin product (LYSOFORTE®). While the concept is scientifically relevant and the experimental framework is well-organized, several concerns have been identified. These issues must be addressed to improve the scientific rigor and clarity of the work prior to further consideration for publication. The comments below are categorized accordingly to assist with revisions.
Major Comments
- In Abstract – Overstatement of Conclusions: The abstract claims that lysolecithin "triggers signaling pathways" and is a "bioactive mixture," which overstates the results from an in vitro model. Rephrase speculative language in the abstract. Consider wording like: "Our findings suggest that lysolecithin may influence gene expression and metabolite transport in vitro, indicating potential regulatory effects."
- Commercial Bias in Language and Product Naming: Repeated mention of LYSOFORTE® brand in abstract, introduction, and discussion implies commercial promotion. It is suggested to limit brand name to one mention in the Materials & Methods. Use generic descriptors (e.g., “soybean-derived lysolecithin product”) in other sections to maintain scientific neutrality.
- Lack of Clear Hypothesis or Study Objective: The manuscript does not provide a focused research hypothesis or clearly defined study aim. It is suggested to add a final paragraph to the Introduction with a statement like: "This study aims to assess whether a soybean-derived lysolecithin affects intestinal epithelial cell viability, gene expression, and metabolite transport in vitro using a triple-assay approach.
- Inadequate Methodological Transparency: The analytical methods for lysolecithin characterization are said to be “confidential.” This undermines reproducibility. Provide at least a high-level summary of critical methodological steps (e.g., retention times, standards used, validation method). If under NDA, clarify exactly which elements are proprietary.
- Justification for Lysolecithin Dose Not Clear: While the 1% concentration showed mild toxicity, the rationale for selecting 0.5% is not clearly explained. Justify the dose selection in the methods.
- RNA-Seq and Metabolomics Analysis Lacks Depth: Details on quality control metrics, normalization, read depth, and mapping are missing. The statistical thresholds used (e.g., padj values, FC cutoffs) are not consistently reported. Expand the RNA-seq methods section to include: read mapping rate, RIN scores, QC thresholds, and how DEGs were filtered. For metabolomics, clarify how metabolite IDs were confirmed (e.g., MS/MS spectral matching, database references).
- Implausibly High Fold Changes in Metabolomics: Reported fold-changes (up to 700x+) are biologically suspect and suggest data processing artifacts or lack of normalization. Recheck data normalization procedures. Consider re-analyzing with log transformation, normalization to internal standards, or applying intensity cutoffs. Use more conservative fold-change thresholds (e.g., 2–10x) for discussion.
- Interpretation of Gene Expression is Overstated: No protein-level validation was provided. The connection between gene expression and functional outcomes is inferred but not shown. Acknowledge in the Discussion that these changes are at the mRNA level only. If possible, add qPCR validation for a few top DEGs (e.g., PDK4, PLIN2) or clearly state this as a limitation.
- Discussion Makes In Vivo Extrapolations from In Vitro Data: The text implies that results from Caco-2 cells are directly applicable to animals or humans which are not. Add a paragraph under “Limitations” acknowledging that Caco-2 cells lack immune and microbiome context, and in vivo trials are needed to confirm these mechanisms.
- Use of Uncharacterized Transcripts in Tables 2 and 3: Many top up/downregulated genes are listed as unannotated (e.g., AC004264.1, AC092756.1) without explanation. Consider removing these from the main discussion and focusing on biologically interpretable genes. Alternatively, perform enrichment/pathway analysis to infer their potential roles.
- Figure Captions and Legends Are Too Brief: Figures such as volcano plots, PCA, and clustering lack full axis labels, statistical threshold annotations, and captions explaining what’s plotted. Revise figure legends to include: Number of replicates, meaning of axis labels, what defines “significant”, Color coding explanations etc.
- Overuse of Enrichment Tools Without Context: GO/KEGG enrichment figures (e.g., Figure 9, Figure 14) are presented without proper contextual discussion. For each key pathway shown, briefly explain its biological relevance in the Results or Discussion (e.g., amino acid transport as a nutritional effect mechanism).
- Lack of Metabolite IDs and Data Provenance: The metabolite tables lack universal identifiers (e.g., KEGG, HMDB, CAS) that allow validation by others. Add a column for each metabolite in Tables 4–6 indicating its KEGG or HMDB ID. Also, mention in Methods how matches were confirmed.
- Inadequate Acknowledgment of Conflict of Interest: The study is fully funded by the producer of the tested compound. While this is disclosed, the manuscript’s tone is still promotional. Include a sentence in the Discussion such as: “Given the commercial involvement in this study, future independent replication is essential to confirm the reported findings.”
- No Clear Statement of Data Availability: The authors refer to data deposited in GEO but do not provide accession numbers. Include the accession number for the RNA-seq dataset and any raw metabolomics data in the Data Availability section.
Minor Comments
- Language and Grammar: Several grammatical errors and overly promotional phrases are present. Revise manuscript for scientific tone and grammatical correctness. Avoid subjective terms like “very potent,” “strongly indicates,” etc.
- Figure and Table Legends: Legends are too brief and often lack information on replicates, statistical thresholds, and axis meanings. Expand legends for clarity. Define all terms and statistical thresholds in each figure/table.
- Inclusion of Uncharacterized Genes in DEGs: Top DEGs include many unannotated transcripts with no biological context. Either remove or explain the potential relevance of these transcripts.
- Inconsistent Use of References: Some references (e.g., #24, #25) are reused repeatedly without deep explanation or critical evaluation. Improve critical engagement with literature. Avoid over-citation of internal references.
- Missing Universal IDs for Metabolites: Tables list metabolite names without KEGG/HMDB IDs. Add universal identifiers to enhance reproducibility and external validation.
- Data Availability Statement: GEO data are referenced, but no accession number is provided. Add RNA-seq and metabolomics data accession numbers or specify submission in progress.
Author Response
Dear reviewer,
We genuinely want to thank you for your thorough revision of our manuscript. All of the comments that you have provided are very relevant, and we truly believe that they helped to increase the quality of the paper and reach the level needed for publication.
First of all, the authors started working on the general comments provided on the manuscript. Main efforts carried out the general text are:
- The authors improved the quality of English language use through external revision using the MDPI English editing service.
- The entire manuscript has been rewritten to make sure that no suggestive sentences were included in the revised manuscript. Furthermore, additional recent references have been added to the manuscript, such that it can be considered more up-to-date.
- The research design has been better described in the reviewed paper, thus increasing its quality.
- The Methods have been completely rewritten based on the requests from the reviewer to improve reproducibility. Extra processes have been included, allowing the reader to better replicate the experiments from scratch.
- The Results have been presented in an amended way, making them clearer to the readers. Also results are more into depth discussed in the discussion section. All method related parts have been removed from the Results section.
- The Conclusions have been better linked to the results. And also, conclusions withdrawn from the in vitro dataset have been softened and it has been clearly stated that there are limitation with regards to interpretation of RNAseq data and also towards extrapolation to the in vivo situation.
Regarding the more specific comments received, we have made the needed changes and provided the reviewers with the associated responses in the following. While the concept is scientifically relevant and the experimental framework is well-organized, several concerns have been identified. These issues must be addressed to improve the scientific rigor and clarity of the work prior to further consideration for publication. The comments below are categorized accordingly to assist with revisions.
Major Comments
#In Abstract – Overstatement of Conclusions: The abstract claims that lysolecithin "triggers signaling pathways" and is a "bioactive mixture," which overstates the results from an in vitro model. Rephrase speculative language in the abstract. Consider wording like: "Our findings suggest that lysolecithin may influence gene expression and metabolite transport in vitro, indicating potential regulatory effects."
→ The authors fully agree with the reviewer’s comment. Therefore, the authors carefully rephrased the assumptions made in the abstract section. “In conclusion, the data on intestinal cell viability, gene expression, and metabolite abundance seem to reveal the bioactivities of lysolecithin. The latter data suggest that the specific lysolecithin source used here is more than a biosurfactant; more specifically, it seems to be a potent bioactive mixture of amphiphilic compounds triggering cell signaling pathways.”
#Commercial Bias in Language and Product Naming: Repeated mention of LYSOFORTE® brand in abstract, introduction, and discussion implies commercial promotion. It is suggested to limit brand name to one mention in the Materials & Methods. Use generic descriptors (e.g., “soybean-derived lysolecithin product”) in other sections to maintain scientific neutrality.
→ The authors fully agree with this approach and have removed the word LYSOFORTE from all sections. Only, as suggested, it is only described in the methods section.
#Lack of Clear Hypothesis or Study Objective: The manuscript does not provide a focused research hypothesis or clearly defined study aim. It is suggested to add a final paragraph to the Introduction with a statement like: "This study aims to assess whether a soybean-derived lysolecithin affects intestinal epithelial cell viability, gene expression, and metabolite transport in vitro using a triple-assay approach.
→ This was a request also raised by the other reviewers and has now been implemented as advised.
#Inadequate Methodological Transparency: The analytical methods for lysolecithin characterization are said to be “confidential.” This undermines reproducibility. Provide at least a high-level summary of critical methodological steps (e.g., retention times, standards used, validation method). If under NDA, clarify exactly which elements are proprietary.
→ The authors understand this concern and have reached out to ITERG to see whether we can provide the readers with extra information. Based on the input of ITERG, this methodology section has been rewritten and updated, so that it improves ease of repeatability. Following your comment, the authors realised that the term confidential was probably inappropriate, as the bibliographical references given in the text provide access to the details of the various analytical methods used. Following your comment, the authors have added a few clarifications to the text, while taking care to remain concise.
#Justification for Lysolecithin Dose Not Clear: While the 1% concentration showed mild toxicity, the rationale for selecting 0.5% is not clearly explained. Justify the dose selection in the methods.
→ The authors included an extra section on the relevance of the dose setting in this study in the methods section. “The in vivo addition of LYSOFORTE in broiler diets is 250g/t. A young broiler chicken eats 200g of feed per day. This implies that the broiler will have an intake of ±0.5g lysolecithin per day. The content of intestinal lumen in the broiler is estimated to be 100ml. So that means 0.5g/100ml = 0.5% exposure of lysolecithin in the intestinal tract per day. Hence, the dosage set up in this in vitro work is physiologically relevant for the in vivo situation.”
#RNA-Seq and Metabolomics Analysis Lacks Depth: Details on quality control metrics, normalization, read depth, and mapping are missing. The statistical thresholds used (e.g., padj values, FC cutoffs) are not consistently reported. Expand the RNA-seq methods section to include: read mapping rate, RIN scores, QC thresholds, and how DEGs were filtered. For metabolomics, clarify how metabolite IDs were confirmed (e.g., MS/MS spectral matching, database references).
→ The authors want to thank the reviewer for this clear input and have rewritten the methods based on these comments. Both the RNAsequencing methodology as well as the metabolomic approach have been rewritten with extra information included.
#Implausibly High Fold Changes in Metabolomics: Reported fold-changes (up to 700x+) are biologically suspect and suggest data processing artifacts or lack of normalization. Recheck data normalization procedures. Consider re-analyzing with log transformation, normalization to internal standards, or applying intensity cutoffs. Use more conservative fold-change thresholds (e.g., 2–10x) for discussion.
→ Indeed, the authors now replotted the data using a cutoff of a FC from 2 up to 20 as threshold for results presentation and subsequent discussion. Therefore also the Tables representing the metabolomic data have been revised.
#Interpretation of Gene Expression is Overstated: No protein-level validation was provided. The connection between gene expression and functional outcomes is inferred but not shown.
→ Indeed the authors acknowledge the limitation that at the intestinal tract level, no protein analyses were performed to screen whether the gene expression changes are effectively being translated into proteins. Factors such as the efficiency of translation (how well mRNA is converted into protein) can significantly impact protein production. Therefore, an extra statement has been added to the results section to pinpoint the limitation in this context.
“The correlation between expression levels of protein and mRNA in mammals is relatively low. Suggested explanations for this low correlation include post-transcriptional regulation and measurement noise. This low correlation makes it difficult to integrate protein and mRNA data. Hence, one needs to be careful when considering the consequences and impact of the RNAseq data presented in the current study.” Perl, K., Ushakov, K., Pozniak, Y. et al. Reduced changes in protein compared to mRNA levels across non-proliferating tissues. BMC Genomics 18, 305 (2017).”
#Discussion Makes In Vivo Extrapolations from In Vitro Data: The text implies that results from Caco-2 cells are directly applicable to animals or humans which are not. Add a paragraph under “Limitations” acknowledging that Caco-2 cells lack immune and microbiome context, and in vivo trials are needed to confirm these mechanisms.
→ The authors completely agree with this request and advise and have included an extra section in the Discussion section confirming that extra studies are needed. Also the conclusion section has been softened so that indeed the need for extra in vivo validation is described.
“The lysolecithin used in this study was fully screened in terms of PL and LPL characteristics, and the authors also tried to use an in vitro dosage that is physiologically relevant, so that the data collected could be potentially extrapolated to the in vivo situation. However, one should acknowledge that Caco-2 cells lack immune and microbiome context, and in vivo trials are needed to confirm these mechanisms.”
“In conclusion, these data suggest that the impact of lysolecithin on intestinal tract cells cannot solely be ascribed to its surface chemistry action. The data presented in this study reveal, for the first time, how PL and LPLs present in lysolecithin can trigger gene expression and intestinal cell function. The gene expression data revealed that the nutrient transporters were upregulated in the lysolecithin-treated intestinal cells. This could potentially facilitate the increased transport of metabolites from the apical lumen (which reflects the intestinal lumen) towards the basolateral compartment (which reflects the bloodstream). Overall, this study seems to reveal that lysolecithin is a mixture of bio-active compounds that can trigger gene expression and metabolism in the intestinal tract. However, extra in vivo data are needed to corroborate the in vitro findings.”
#Use of Uncharacterized Transcripts in Tables 2 and 3: Many top up/downregulated genes are listed as unannotated (e.g., AC004264.1, AC092756.1) without explanation. Consider removing these from the main discussion and focusing on biologically interpretable genes. Alternatively, perform enrichment/pathway analysis to infer their potential roles.
→ The authors did take this advice and removed the unannotated genes from the Results and discussion. The list has now been updated with all genes that can be linked to clear functions in case they would be translated into proteins.
#Figure Captions and Legends Are Too Brief: Figures such as volcano plots, PCA, and clustering lack full axis labels, statistical threshold annotations, and captions explaining what’s plotted. Revise figure legends to include: Number of replicates, meaning of axis labels, what defines “significant”, Color coding explanations etc.
→ The authors have tried as much as possible to address this concern by updating all Figure legends keeping this request into consideration.
#Overuse of Enrichment Tools Without Context: GO/KEGG enrichment figures (e.g., Figure 9, Figure 14) are presented without proper contextual discussion. For each key pathway shown, briefly explain its biological relevance in the Results or Discussion (e.g., amino acid transport as a nutritional effect mechanism).
→ The authors have tried to address this and added for Figure 11, Figure 15 and Figure 19 an extra section into the discussion of the paper. “However, when looking more into depth in Figure 11, Figure 15 and Figure 19, it becomes more and more clear that in the intestinal tract, amino acid transporters and genes related to lipid metabolism are increasingly expressed in the lysolecithin treatment versus controls. By contrast in the metabolite profile pathways, one can see more prominently the increased lipid metabolite pathways being affected. This is very interesting and confirms previous in vivo work [54].”
#Lack of Metabolite IDs and Data Provenance: The metabolite tables lack universal identifiers (e.g., KEGG, HMDB, CAS) that allow validation by others. Add a column for each metabolite in Tables 4–6 indicating its KEGG or HMDB ID. Also, mention in Methods how matches were confirmed.
→ This precious comment has been tackled by the authors. All Tables have been remade with this in mind.
#Inadequate Acknowledgment of Conflict of Interest: The study is fully funded by the producer of the tested compound. While this is disclosed, the manuscript’s tone is still promotional. Include a sentence in the Discussion such as: “Given the commercial involvement in this study, future independent replication is essential to confirm the reported findings.”
→ The authors acknowledge this concern and have added an extra sentence in the discussion.
#No Clear Statement of Data Availability: The authors refer to data deposited in GEO but do not provide accession numbers. Include the accession number for the RNA-seq dataset and any raw metabolomics data in the Data Availability section.
→ The authors must state that the submission of the FASTA files is in progress at SRA but we encounter some platform updates due which we could not progress; submission name is SUB15426483. The authors have sent an email to the SRA contact in order to request for help. The authors promise they will persist and make all raw data-files publicly available asap.
#Language and Grammar: Several grammatical errors and overly promotional phrases are present. Revise manuscript for scientific tone and grammatical correctness. Avoid subjective terms like “very potent,” “strongly indicates,” etc.
→ The authors have removed as much as possible the subjective terms as indeed this is very important. Also, the authors have submitted the manuscript meanwhile for English editing via the MDPI services.
#Figure and Table Legends: Legends are too brief and often lack information on replicates, statistical thresholds, and axis meanings. Expand legends for clarity. Define all terms and statistical thresholds in each figure/table.
→ The authors have tried to address this as much as possible. We thoroughly believe it improves the quality of the paper.
#Inclusion of Uncharacterized Genes in DEGs: Top DEGs include many unannotated transcripts with no biological context. Either remove or explain the potential relevance of these transcripts.
→ These transcripts have been removed by the authors in the revised manuscript. Of course, when completing the GEO deposition, all possible annotations will be available for the readers. But for now, only the genes with different expressions that can be linked to a biological function have been presented in the Tables to the readers.
#Inconsistent Use of References: Some references (e.g., #24, #25) are reused repeatedly without deep explanation or critical evaluation. Improve critical engagement with literature. Avoid over-citation of internal references.
→ The authors have been looking into this topic and removed the use of multiple citations of the same source. We hope this solves the request from the reviewer.
#Missing Universal IDs for Metabolites: Tables list metabolite names without KEGG/HMDB IDs. Add universal identifiers to enhance reproducibility and external validation.
→ This has been addressed carefully by the authors. All genes and metabolites now have the universal identifiers linked to them.
#Data Availability Statement: GEO data are referenced, but no accession number is provided. Add RNA-seq and metabolomics data accession numbers or specify submission in progress.
→ Submission is indeed in progress.
Reviewer 3 Report
Comments and Suggestions for Authors
This manuscript explores the gene regulatory and metabolic potential of a lysolecithin-based product (LYSOFORTE®) in Caco-2 cells using a multi-modal approach, including viability assays, RNA sequencing, and untargeted metabolomics. The topic is timely and relevant to both molecular biology and animal nutrition, particularly in understanding bioactive roles of feed additives. The overall structure is clear, and the data presentation is extensive; however, there are several major issues that must be addressed before the manuscript is suitable for publication.
The study is fully funded and conducted by Kemin Europa N.V., the manufacturer of LYSOFORTE®. Although this is disclosed, the manuscript would benefit from a clearer statement regarding data integrity and scientific independence to ensure transparency and credibility. The composition analysis of the lysolecithin product is incomplete, as more than half of the mixture (58.5%) remains uncharacterized. The authors should elaborate on the possible constituents of this uncharacterized fraction and how they may contribute to the observed biological effects.
The concentration of lysolecithin used in the experiments (0.5%) is chosen based on MTT viability screening, yet its physiological relevance remains uncertain. It is important to discuss whether such concentrations are realistically encountered in the intestinal lumen following dietary supplementation. Without this context, it is difficult to evaluate the translational potential of the findings.
The RNA-Seq data reveal differential expression of several hundred genes; however, there is no follow-up validation of key transcripts, and no protein-level data are provided. A discussion of this limitation is necessary, particularly given that mRNA changes do not always correlate with functional outcomes. Similarly, the metabolomics analysis presents extensive lists of altered metabolites, but interpretation remains largely descriptive. The manuscript would be strengthened by focusing on a subset of relevant metabolites and explaining their potential physiological or mechanistic significance, especially in connection with the gene expression findings.
The manuscript draws broad conclusions about the in vivo implications of lysolecithin based solely on in vitro data. The discussion should better distinguish between in vitro observations and their potential in vivo relevance, and temper claims accordingly.
The figure quality, particularly resolution and labeling, is suboptimal in several places and should be improved to meet publication standards. Additionally, red and green color schemes used in heatmaps are not color-blind friendly. More accessible color palettes should be adopted to ensure readability by all readers. A number of minor typographical errors and awkwardly phrased sentences are present throughout the manuscript, and a thorough proofreading is needed to ensure clarity and consistency.
Author Response
Dear reviewer,
We genuinely want to thank you for your thorough revision of our manuscript. All of the comments that you have provided are very relevant, and we truly believe that they helped to increase the quality of the paper and reach the level needed for publication.
First of all, the authors started working on the general comments provided on the manuscript. Main efforts carried out the general text are:
- The authors improved the quality of English language use through external revision using the MDPI English editing service.
- The entire manuscript has been rewritten to make sure that no suggestive sentences were included in the revised manuscript. Furthermore, additional recent references have been added to the manuscript, such that it can be considered more up-to-date.
- The research design has been better described in the reviewed paper, thus increasing its quality.
- The Methods have been completely rewritten based on the requests from the reviewer to improve reproducibility. Extra processes have been included, allowing the reader to better replicate the experiments from scratch.
- The Results have been presented in an amended way, making them clearer to the readers. Also results are more into depth discussed in the discussion section. All method related parts have been removed from the Results section.
- The Conclusions have been better linked to the results. And also, conclusions withdrawn from the in vitro dataset have been softened and it has been clearly stated that there are limitation with regards to interpretation of RNAseq data and also towards extrapolation to the in vivo situation.
Regarding the more specific comments received, we have made the needed changes and provided the
Revisions here one-by-one:
#The study is fully funded and conducted by Kemin Europa N.V., the manufacturer of LYSOFORTE®. Although this is disclosed, the manuscript would benefit from a clearer statement regarding data integrity and scientific independence to ensure transparency and credibility.
→ The authors fully agree with this request and have added a sentence to the discussion sections that states: “Hence, one needs to be careful when considering the consequences and impact of the intestinal tract transcriptome data presented in the current study. Also, given the commercial involvement in this study, future independent replication is essential to confirm the reported findings.”
#The composition analysis of the lysolecithin product is incomplete, as more than half of the mixture (58.5%) remains uncharacterized. The authors should elaborate on the possible constituents of this uncharacterized fraction and how they may contribute to the observed biological effects.
→ The authors acknowledge that that the remaining part of the lysolecithin is lipid soluble and is removed and remains in the 0.2µm filter after processing the culture medium for the cells because it is not water soluble. It are the amphiphilic and hydrophilic substances that remain in the culture medium that is brought in contact with the cells. Here the PL and LPL are the most important fractions. Hence, the authors believe that the triglyceride and other oily fractions do not have any impact on the cell cultured in the well dishes.
#The concentration of lysolecithin used in the experiments (0.5%) is chosen based on MTT viability screening, yet its physiological relevance remains uncertain. It is important to discuss whether such concentrations are realistically encountered in the intestinal lumen following dietary supplementation. Without this context, it is difficult to evaluate the translational potential of the findings.
→ The authors believe this is a very solid remark and have included an extra section in the methods part to show the calculations behind the proposal that it is a physiologically relevant dosage used in this in vitro setup. “Note: The in vivo addition of lysolecithin in broiler diets is 250g. A young broiler chicken eats 200g of feed per day. This implies that the broiler will have an intake of ±0.5g lysolecithin per day. The content of intestinal lumen in the broiler is estimated to be 100ml. So that means 0.5g/100ml = 0.5% exposure of lysolecithin in the intestinal tract per day. Hence, the dosage set up in this in vitro work is physiologically relevant for the in vivo situation.”
#The RNA-Seq data reveal differential expression of several hundred genes; however, there is no follow-up validation of key transcripts, and no protein-level data are provided. A discussion of this limitation is necessary, particularly given that mRNA changes do not always correlate with functional outcomes.
→ This is indeed, again, a very valid remark. Therefore, the authors have included an extra section in the discussion part of the paper to pinpoint towards this limitation. “It is important to keep in mind, when interpreting the RNAsequencing data, that the correlation between expression levels of protein and mRNA in mammals is relatively low [56]. Suggested explanations for this low correlation include post-transcriptional regulation and measurement noise. This low correlation makes it difficult to integrate protein and mRNA data [56]. Hence, one needs to be careful when considering the consequences and impact of the intestinal tract transcriptome data presented in the current study. Also, given the commercial involvement in this study, future independent replication is essential to confirm the reported findings.”
#Similarly, the metabolomics analysis presents extensive lists of altered metabolites, but interpretation remains largely descriptive. The manuscript would be strengthened by focusing on a subset of relevant metabolites and explaining their potential physiological or mechanistic significance, especially in connection with the gene expression findings.
→ The authors have included an extra statement by focusing in the discussion on the enriched pathways presented in Figure 9, 13, 17. Here the link between transcriptomics and metabolomics has been made.
#The manuscript draws broad conclusions about the in vivo implications of lysolecithin based solely on in vitro data. The discussion should better distinguish between in vitro observations and their potential in vivo relevance, and temper claims accordingly.
→ Indeed, to soften this, extra statements have been made in the discussion and conclusion sections so that the reader is aware that extra in vivo trials are needed to validate the in vivo relevance for the animal.
#The figure quality, particularly resolution and labeling, is suboptimal in several places and should be improved to meet publication standards. Additionally, red and green color schemes used in heatmaps are not color-blind friendly. More accessible color palettes should be adopted to ensure readability by all readers.
→ The authors have done their utter best to tackle this issue by including extra text in the figures stating whether it concerns up and/or down-regulations. Also in the Figure tables, extra efforts have been made to clarify what is presented in the graphs. Also the contrast of the images has been improved so that it should suit/meet requirements for publication.
#A number of minor typographical errors and awkwardly phrased sentences are present throughout the manuscript, and a thorough proofreading is needed to ensure clarity and consistency.
→ With this in mind, the paper has been reviewed by MDPI services by native English speakers and we have a certificate for this so this can be provided upon request. We will also proactively upload this with the resubmission of the manuscript.
Round 2
Reviewer 1 Report
Comments and Suggestions for Authors
I have no other concerns.
Reviewer 3 Report
Comments and Suggestions for Authors
The authors have successfully addressed all of the reviewer’s concerns and questions.